



# Investigating small ion number size distributions:
# insight into cluster formation and growth
Santeri Tuovinen[1], Janne Lampilahti[1], Nina Sarnela[1], Chengfeng Liu[1], Yongchun Liu[2] and Markku
Kulmala[1,2] and Veli-Matti Kerminen[1]
[1]Institute for Atmospheric and Earth System Research/Physics, Faculty of Science, University of
Helsinki, Helsinki, Finland
[2]Aerosol and Haze Laboratory, Beijing Advanced Innovation Center for Soft Matter Science and
Engineering, Beijing University of Chemical Technology, Beijing, China
**Correspondence:** Santeri Tuovinen (santeri.tuovinen@helsinki.fi) and Markku Kulmala
(markku.kulmala@helsinki.fi)

## Abstract

Small ions, consisting mostly of charged molecular clusters with mobility diameters below 2 nm,
exist continuously in the atmosphere. Here, we studied  small ion number size distributions
measured with Neutral cluster and Air Ion Spectrometer measurements in Hyytiälä, Finland and
Beijing, China. We found that in Hyytiälä, there is a strong positive relationship between the
concentration and diameter of small ions of both polarities and highly oxidized organic molecule
(HOM) and sulfuric acid concentrations, and that the relationship with the former is especially
strong. The relationship between the negative sulfuric acid cluster ions and the small ion number
size distribution in Hyytiälä was found to be more complex, but overall positive. In contrast to
Hyytiälä, we found that in Beijing the small ion number size distribution does not have a clear
relationship with sulfuric acid or oxidized organic molecule (OOM) concentration. However, in
both locations, the impact of growth on the small ion number size distribution during periods of
intense cluster formation and new particle formation is clearly seen.

## 1 Introduction

Atmospheric aerosol particles influence the Earth's climate (e.g., Quaas et al., 2009; Boucher et al.,
2013; Schmale et al., 2021; Li et al., 2022) and can have adverse effects on human health (e.g.,
Shiraiwa et al., 2017; Arfin et al., 2023). These influences have commonly been related to
properties, such as the mass or number concentration of an atmospheric aerosol population, its size
distribution, or its chemical composition (Shiraiwa et al., 2017; Atkinson et al., 2015; Finlay, 2021).
The electric charging state of atmospheric aerosols has attracted much less interest, although this
property may have large influences on the dynamics of atmospheric aerosol populations (Harrison
and Carslaw, 2003; Fdez-Arroyabe et al., 2022), thereby affecting many other important aerosol
properties. The presence of charges also makes it possible to measure low aerosol concentrations at
high resolution in both time and particle size (Mirme and Mirme, 2013; Mirme et al., 2024).
Charged atmospheric particles, or more broadly ions, include charged aerosol particles, charged
molecular clusters, and even large molecules having a charge. Ions with electrical mobility
diameters roughly below 2 nm in diameter are classified as small ions, and consist of charged
molecular clusters, while ions above 2 nm consist of charged aerosol particles (Tammet, 1995; Ehn



et al., 2010). Of these charged aerosol particles, those with diameters between 2 and 7 nm are
referred to as intermediate ions (Tammet, 1995).
Atmospheric ions are created through ionization of molecules the atmosphere. Most important of
these ionization sources are cosmic ray radiation, gamma radiation, and radon decay (Harrison and
Tammet, 2008). Small ions are constantly present in the troposphere as molecules are ionized and
subsequently grow to small ions (Harrison and Tammet, 2008; Hirsikko et al., 2011). The lifetime of
small ions is short at around 100 s, and their chemical composition depends on the atmospheric
trace gas concentrations and their chemistry (Harrison and Tammet, 2008; Ehn et al., 2010; Shuman
et al., 2015). In contrast, intermediate ions are typically detected mainly during the occurrence of
atmospheric new particle formation (Tammet et al., 2014, Tuovinen et al., 2024), or during snowfall
or rain (Hirsikko et al., 2007; Tammet et al., 2014). New particle formation (NPF) is considered to
occur when constantly existing stable clusters, neutral or charged, start to grow to larger sizes by
uptake of precursor vapors such as sulfuric acid and organic compounds with low volatilities
(Kulmala et al., 2006; Kulmala et al., 2007; Lehtipalo et al., 2018; Kirkby et al., 2023).
A recent study by Kulmala et al. (2024a) presented the use of a novel cluster ion counter (CIC) for
measuring small and intermediate ion concentrations to study local-scale NPF and to derive other
parameters such as condensation sink (CS). The information gained by these measurements can be
used further to study the complex climate-biosphere feedbacks (Kulmala et al., 2020; Kulmala et
al., Kulmala et al., 2024b). These recent advances have motivated us to take a deeper look at the
small ion size distribution.
The concentration of small ions depends on the ionization rate and the losses of small ions due to
ion-ion recombination, coagulation with larger aerosol particles, and deposition (Tammet et al.,
2006; Hõrrak et al., 2008). The size of small ions depends on their chemical composition and age as
the ions grow through chemical reactions and condensation of vapors, or through coagulation with
neutral clusters. By investigating small ion number size distributions, we can learn more about these
chemical and dynamical processes.
In this study, we combine ion number size distribution data measured by Neutral cluster and Air Ion
Spectrometer  (NAIS; Manninen et al., 2009; Mirme and Mirme, 2013) with concentrations of low-
volatility vapors and ion clusters measured by mass spectrometer instruments to identify how, and
why, the size distribution of small ions changes and evolves. Data from two different contrasting
locations, Hyytiälä, Finland and Beijing, China (Kulmala et al., 2025), are used. First, we will study
if the variation of the small ion size distribution with season is considerable. Secondly, we will
quantify the potential relationship of organic low-volatility vapors and sulfuric acid on the size and
number of small ions. Thirdly, we will analyze the small ion size distribution as a function of
intensity of NPF to reveal how the small ion size distribution changes as the clusters grow. Finally,
some case studies are presented. With these, we aim to identify the most important processes
impacting the small ion number size distribution, and to evaluate the role of these processes in
driving the growth of small ions to intermediate ions.



## 2 Background and methods

### 2.1 Evolution of small ion size distribution

Typically, the parameter of interest when considering small ions is their total number concentration
and its temporal evolution. The changing in the small ion number concentration can be described by
the simplified air ion balance equation:

$$\frac{dN^\pm}{dt} = Q - CoagS\, N^\pm - \alpha\, N^\pm N^\mp - S\, N^\pm \tag{1}$$

Here, $N^\pm$ is the concentration of one polarity, while $N^\mp$ is the concentration of the other polarity.
The first term on the right-hand side of the equation describes the source rate of the ions, where $Q$ is
the ionization rate of air molecules. The second term, where CoagS stands for coagulation sink,
tells the loss rate of small ions due to coagulation on larger aerosol particles. The third term tells the
loss rate of ions due to ion-ion recombination, where $\alpha$ is the ion recombination coefficient. The
final term describes other losses of the ions, including deposition, and $S$ is the loss rate of the ions to
these other sinks.

As we can see, the above equation does not explicitly depend on the size of the small ions nor can it
be directly used to describe the evolution of the size-dependent small ion size distribution. The time
evolution of small ions of certain size $i$ are described by the charged general dynamics equations
(charged GDEs; Kulmala et al., 2012):

$$\frac{d\,N_i^\pm}{dt} = J_i + \chi\, N_i N_{d<i}^\pm - N_i^\pm\, CoagS_i - \alpha\, N_i^\pm N_{d<i}^\mp - \frac{GR}{\Delta d_i} N_i^\pm \tag{2}$$

Here, $J_i$ is the formation rates of ions of size $i$. The second term on the right-hand side represents the
charging of neutral clusters by ions smaller than $i$, where $X$ is the ion-cluster attachment coefficient.
The last term, where GR is the ion growth rate, describes the growth of ions $i$ to larger sizes.
Considering Eq. 2, we can see that an increasing GR will shift the ion size distribution towards
larger diameters. CoagS is the highest for the smallest ions and if it increases, the concentrations of
smallest ions are decreasing the most, causing an apparent shift in the distribution towards larger
diameters. However, CoagS also affect the lifetime of small ions, so that with an increasing CoagS
the ions have less time to grow, reducing the concentration of larger small ions. If ion
concentrations are high, ion-ion recombination rate will be higher, which will also lead to shorter
small ion lifetime and possibly smaller concentrations of larger small ions. Through ion-cluster
attachment, the small ion size distribution depends on the size distribution of neutral clusters,
although this term is relatively small when compared to the coagulation loss and growth terms. In
this study, we are mainly interested in the formation and growth of ions, and their impacts on the
small ion size distribution. Impacts of coagulation scavenging or ion-ion recombination on the small
ion size distribution are not explicitly considered in this study.

### 2.2 Measurement sites

Two different locations were considered in this study: SMEAR II measurement station in Hyytiälä,
Finland (61°51′ N, 24°17′ E) and BUCT/AHL measurement station in Beijing, China (39°94′ N,
116°30′ E). The former is a rural site surrounded by boreal forest while the latter is an urban site
close to residential building and traffic roads. For more details on SMEAR II station, see Hari and
Kulmala. (2005). For more details on BUCT/AHL site, see Liu et al. (2020). These two locations





are included in the analysis due to their contrasting natures, providing an opportunity for insight
into the variation of small ion size distribution and small ion dynamics in different environments.
**2.3 Measurement and other data**

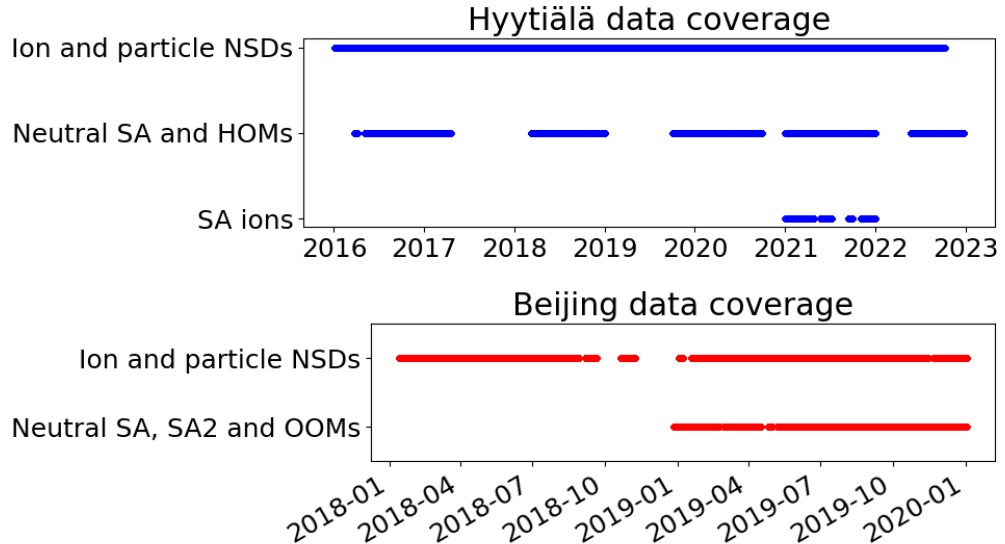

**Fig.1:** Data coverage for the two sites, Hyytiälä, Finland, and Beijing, China, from which data was
used in this study. NSD refers to number size distribution, while SA refers to sulfuric acid, SA2 to
neutral sulfuric acid dimer, HOM to highly oxidized organic molecule and OOM to oxidized
organic molecule.
Atmospheric ion and total particle number size distributions in Hyytiälä and Beijing were measured
with Neutral cluster and Air Ion Spectrometer (NAIS; Manninen et al., 2009; Mirme and Mirme,
2013). The NAIS measures both charged and total particle number size distributions in the ranges
0.8-42 nm and 2.5–42 nm, respectively. Main focus of the analysis in this study is on the number
size distributions of small ions (diameters below 2 nm). Ion concentrations between 2.0 and 2.3 nm
were used to characterize the intensity of local clustering (Tuovinen et al., 2023) and new particle
formation ranking data, characterizing the intensity of NPF, was also used. The NPF ranking was
based on the total particle number concentration between 2.5 and 5 nm and determined according to
the method presented by Aliaga et al. (2023).
All diameters used in study are electrical mobility diameters. We note that especially for the
smallest of the ions the mobility diameter may not accurately describe the physical dimensions of
the ion (see e.g., Ehn et al., 2011). Regardless, we refer to diameter rather than electrical mobility as
we see it as more intuitively understandable parameter for the ion size.
From Hyytiälä, concentrations of neutral sulfuric acid and highly oxidized organic molecules
(HOMs) were used to study the influence of cluster formation and growth on the small ion size
distribution. These were measured with Chemical Ionization Atmospheric Pressure interface Time-
Of-Flight (CI-APi-TOF) mass spectrometer (Jokinen et al., 2012). In addition, the signal counts of
ionized sulfuric acid clusters measured with APi-TOF were used to give further insight into the

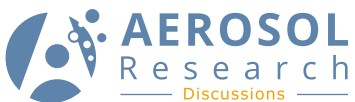

composition of the small ions. The signal counts in the study are given as relative signals to the total
measured ion current. From Beijing, neutral sulfuric acid, sulfuric acid dimer and total oxidized
organic molecule (OOM) concentrations, which were measured with a nitrate based – long time-of-
flight chemical ionization mass spectrometer (CIMS), were included in the analysis. We note that
we use the term OOM instead of HOM for the organic molecules in Beijing based on previous
results by Yan et al. (2021), suggesting that most of these measured organic molecules in Beijing do
not meet the requirements for HOMs (see Bianchi et al., 2019).
Data coverage for both sites is presented in Fig. 1.

## 2.4 Determining the average small ion diameter

From the small ion number size distributions, we determined the mean mobility diameter ($d_{mean}$),
and median mobility diameter ($d_{median}$) of small ions. First, cubic interpolation was applied to the
measured ion number size distributions. We note that nearest neighbor and linear interpolation
methods were also tested, and the influence of the chosen method on the value of $d_{mean}$ or $d_{median}$ was
found minor. The diameter range for the interpolation was from the lower detection limit to 2 nm
with a step of 0.001 nm. Then, $d_{peak}$ was determined by finding the diameter corresponding to the
maximum concentration of small ions. Weighted mean and median were used to determine $d_{mean}$ and
$d_{median}$, with the number concentrations of ions below 2 nm in diameter used as weights. The
equation below was used to find weighted mean diameter:

$$d_{mean} = \frac{\sum N_i d_i}{\sum N_i}, \tag{3}$$

where $d_i$ is the diameter of ions of a certain size and $N_i$ is their number concentration. The weighted
median was determined by finding the diameter $d_j$ satisfying

$$j = \min_k \left[ \sum N_i d_i > \frac{1}{2} \sum N_i d_i \right]. \tag{4}$$

## 3 Results

### 3.1 Seasonal variation of the small ion size distribution

#### *3.1.1 Hyytiälä*

The upper panel of Fig. 2 shows the monthly median negative and positive ion distributions
between 0.8 and 2 nm in Hyytiälä. Table 2 records the monthly mean and median diameters ($d_{mean}$
and $d_{median}$). Clear  month-to-month changes in the size distributions are observed, and these are
more pronounced for negative ions. During winter, the concentration of negative ions peaks already
below 1 nm, while during summer the highest concentration is between 1.1 and 1.2 nm. In addition,
the concentrations of negative ions above 1.1 nm are increased from winter to summer. Close to 2
nm, the ion concentration is almost one order of magnitude higher during summer. This behavior of
the size distribution is reflected in the values of $d_{mean}$ and $d_{median}$, which are smallest  during
December and January, with $d_{mean}$ = 0.99 nm and $d_{median}$ = 0.95 nm, and the largest during June and
July, with $d_{mean}$ = 1.15 nm and $d_{median}$ = 1.11 nm.
Positive ion size distributions behave similarly as the negative ones, however the changes are less
pronounced. Above 1.4 nm and up to 2 nm, the concentrations are roughly twice as high, or less,

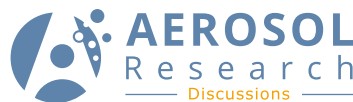

during summer compared to winter. For positive small ions, the smallest value of $d_{\mathrm{mean}} = 1.16$ and
$d_{\mathrm{median}} = 1.13$ nm (December and January) and the largest value of $d_{\mathrm{mean}} = 1.29$ nm and $d_{\mathrm{median}} = 1.27$
nm (June). The difference between the average diameters of negative and positive small ions was
around 0.15 nm, in line with previous studies (e.g., Hõrrak et al., 2000).
The observed seasonal behavior of the size distributions in Hyytiälä follow expectations: during
spring and summer, the concentrations of low-volatility vapors are much higher due to increased
solar radiation and organic emissions (Sulo et al., 2021). Therefore, small ions should be able to
grow to larger diameters due to the uptake of these vapors. We will look further into how the small
ion size distribution varies with respect to low volatile vapor concentrations in Sect. 3.2.
### 3.1.2 Beijing

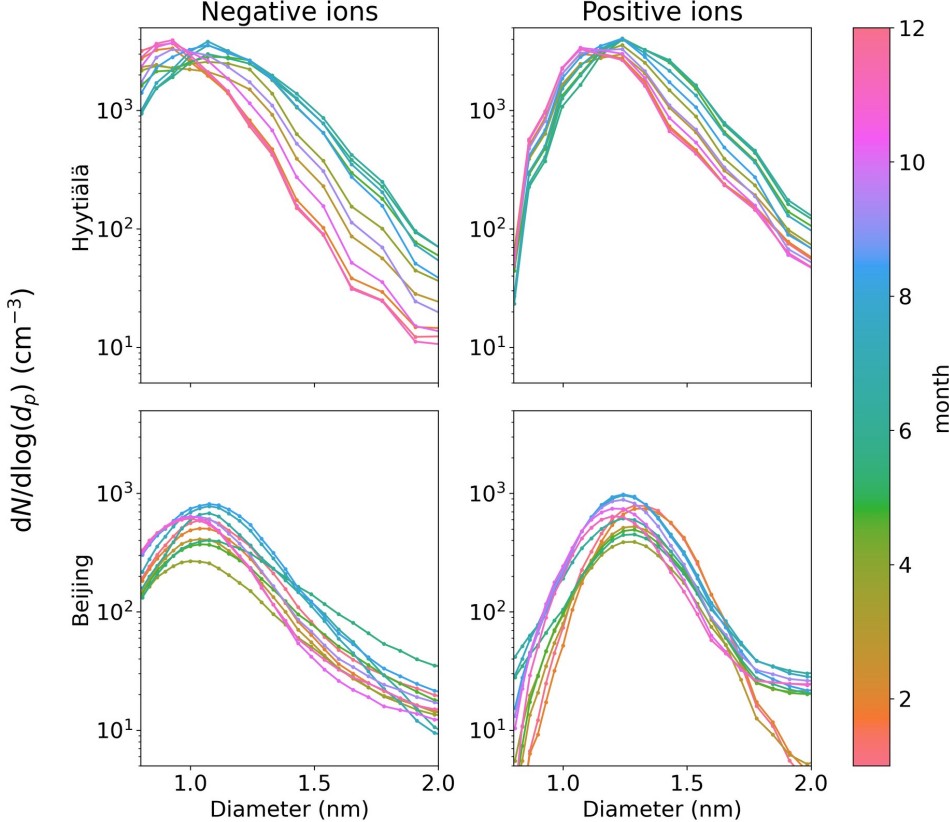

**Fig. 2**: Median monthly sub-2 nm negative (left) and positive (right) ion size distributions for
Hyytiälä (top) and Beijing (bottom). The different months are marked by different colors.
The bottom panel of Fig. 2 shows the monthly median negative and positive ion distributions
between 0.8 and 2 nm in Beijing, while Table 3 records the monthly $d_{\mathrm{mean}}$ and $d_{\mathrm{median}}$. Compared to
Hyytiälä, the seasonal trends in Beijing are much more unclear and complex. For negative ions, the
concentrations during summer are higher than in other seasons below 1.6 nm and lower than in
other seasons close to 2 nm. During spring, the concentrations of negative ions below 1.4 nm are



lower than in other seasons. The smallest value of negative $d_{mean}$ and $d_{median}$ are in November, $d_{mean}$ = 1.04 nm and $d_{median}$ = 1.01 nm. The largest values are during June, $d_{mean}$ = 1.16 nm and $d_{median}$ = 1.12 nm.

For positive small ions in Beijing, the concentrations of ions close to 0.8 nm and 2 nm are both considerably lower from January to March compared to later months. Otherwise, it is difficult to identify any clear patterns. The largest positive average diameter is during February, $d_{mean}$ = 1.32 nm and $d_{median}$ = 1.31 nm, while the smallest values are in November, $d_{mean}$ = 1.22 nm and $d_{median}$ = 1.21 nm.

We note that because there are less data from Beijing compared to Hyytiälä, variation between years can have larger impact on the results than in Hyytiälä.

**Table 1:** Mean and median monthly diameters (nm) of ions between 0.8 and 2 nm in Hyytiälä. *The highest concentration corresponds to the lowest detected diameter.

| | Negative ions | | Positive ions | |
|---|---|---|---|---|
| Month | $d_{mean}$ | $d_{median}$ | $d_{mean}$ | $d_{median}$ |
| 1 | 0.99 | 0.95 | 1.16 | 1.13 |
| 2 | 1.00 | 0.96 | 1.17 | 1.15 |
| 3 | 1.05 | 1.02 | 1.2 | 1.18 |
| 4 | 1.08 | 1.06 | 1.21 | 1.20 |
| 5 | 1.12 | 1.10 | 1.27 | 1.25 |
| 6 | 1.15 | 1.11 | 1.29 | 1.27 |
| 7 | 1.15 | 1.11 | 1.28 | 1.26 |
| 8 | 1.13 | 1.10 | 1.25 | 1.23 |
| 9 | 1.11 | 1.08 | 1.22 | 1.20 |
| 10 | 1.05 | 1.02 | 1.19 | 1.17 |
| 11 | 1.01 | 0.98 | 1.17 | 1.14 |
| 12 | 0.99 | 0.95 | 1.16 | 1.13 |

**Table 2:** Mean and median monthly diameters (nm) of ions between 0.8 and 2 nm in Beijing. *The highest concentration corresponds to the lowest detected diameter.

| | Negative ions | | Positive ions | |
|---|---|---|---|---|
| Month | $d_{mean}$ | $d_{median}$ | $d_{mean}$ | $d_{median}$ |
| 1 | 1.10 | 1.07 | 1.30 | 1.30 |
| 2 | 1.09 | 1.06 | 1.32 | 1.31 |
| 3 | 1.09 | 1.06 | 1.28 | 1.27 |
| 4 | 1.10 | 1.05 | 1.28 | 1.26 |



| 5 | 1.12 | 1.07 | 1.28 | 1.27 |
| 6 | 1.16 | 1.12 | 1.27 | 1.26 |
| 7 | 1.12 | 1.08 | 1.25 | 1.24 |
| 8 | 1.10 | 1.08 | 1.25 | 1.24 |
| 9 | 1.10 | 1.07 | 1.25 | 1.24 |
| 10 | 1.07 | 1.03 | 1.24 | 1.23 |
| 11 | 1.04 | 1.01 | 1.22 | 1.21 |
| 12 | 1.05 | 1.01 | 1.23 | 1.21 |

218

## 3.2 Potential impact of low volatility vapors to small ion size distribution in Hyytiälä

### 3.2.1 highly oxidized organic molecules (HOMs)

Fig. 3 shows the median ion number size distributions between 0.8 and 2 nm in Hyytiälä with respect to varying neutral highly oxidized organic molecule (HOM) concentration. HOM monomers, HOM dimers and total HOM are considered separately. Results for daytime (10:00-16:00) and evening (18:00-00:00) are both presented (Fig. 3a and 3b, respectively). The HOM concentrations are divided into percentiles.

A clear increase in the number of negative ions above approx. 1.05 nm, and for positive ions slightly larger than that, is seen with an increasing HOM concentration percentile for all the plotted HOM categories. The difference is largest for HOM monomers and HOM total, which is mainly dominated by the HOM monomers. The difference is also stronger for negative ions than positive ions, and is stronger during the evening (Fig. 3b) compared to daytime (Fig. 3a).

Comparing the negative ion size distributions between the HOM percentiles of 0-20% and 80-100%, we see that the difference in the concentrations increases with an increasing diameter, and that close to 2 nm this difference is approximately one order of magnitude during daytime (Fig. 3a) and a bit more than that during the evening (Fig. 3b). The negative ion size distributions for different HOM percentiles are otherwise quite similar during the daytime and evening, however during the evening the difference between the respective size distributions for HOM monomer percentile 60-80% and 80-100% is higher (Fig. 3b). During daytime, the ion concentrations are similar in the 60-80% and 80-100% percentiles (Fig. 3a), while during the evening, the concentration close to 2 nm is around twice as high when HOM monomer concentration is in the 80-100% percentile compared to 60-80% percentile (Fig. 3b). Comparing similar negative ion concentrations when HOM monomer concentration during the evening is in the 80-100% percentile compared to 0-20%, there's approx. a 0.5 nm shift in diameters, a major difference for the sub-2 nm ion population.

In line with the large differences in the small ion size distributions in Fig. 3 with respect to HOM concentration, a strong correlation between the small ion $d_{mean}$ and the HOM concentrations was seen (Fig. 4). The Spearman correlation coefficients ($r_s$) between $d_{mean}$ and HOMs were 0.6 or above, for both daytime and evening. For daytime, the best correlation was between $d_{mean}$ of positive



ions and HOM monomer concentration, $r_s = 0.74$. During nighttime, the strongest correlation was
between $d_{mean}$ of negative ions and HOM monomer concentration, $r_s = 0.73$. By using a variable
such as $d_{mean}$, we have been able to get some insight about the behavior of the underlying ion size
distribution.
The clear correlation between HOMs and the small ion size distribution in Hyytiälä suggests a
strong impact of organic compounds to the small ion population. This interpretation, as opposed to
the correlation being due to a correlation with another variable such sulfuric acid concentration, is
supported by the observation of the correlation being stronger during evening when the
concentrations of other precursors such as sulfuric acid are lower and organic ion cluster formation
is known to take place in Hyytiälä (Mazon et al., 2016; Rose et al., 2018). Part of the increase in
diameters of small ions when HOMs are abundant could be due to the large size of organic
molecules when compared to sulfuric acid molecules. However, the clear increase in concentrations
even close to 2 nm suggests that a significant part of the impact is due to the growth of small ions
by uptake of organic vapors.
While the concentrations of larger small ions of both polarities increase with increasing HOM
percentile, the differences are larger for the negative ions (Fig. 3). This could be due to the uptake
of organics being more effective for negatively charged ions. However, the equally strong
correlation between $d_{mean}$ of positive small ions and HOM concentrations does not support this
interpretation. A possible explanation is the size difference between the negative and positive small
ions: due to the larger diameter of positive small ions, it might be that the impact of the growth to
the diameters of the positive ions is not as large. For a larger cluster (ion), more molecules are
needed to increase the diameter equally than for a smaller one.



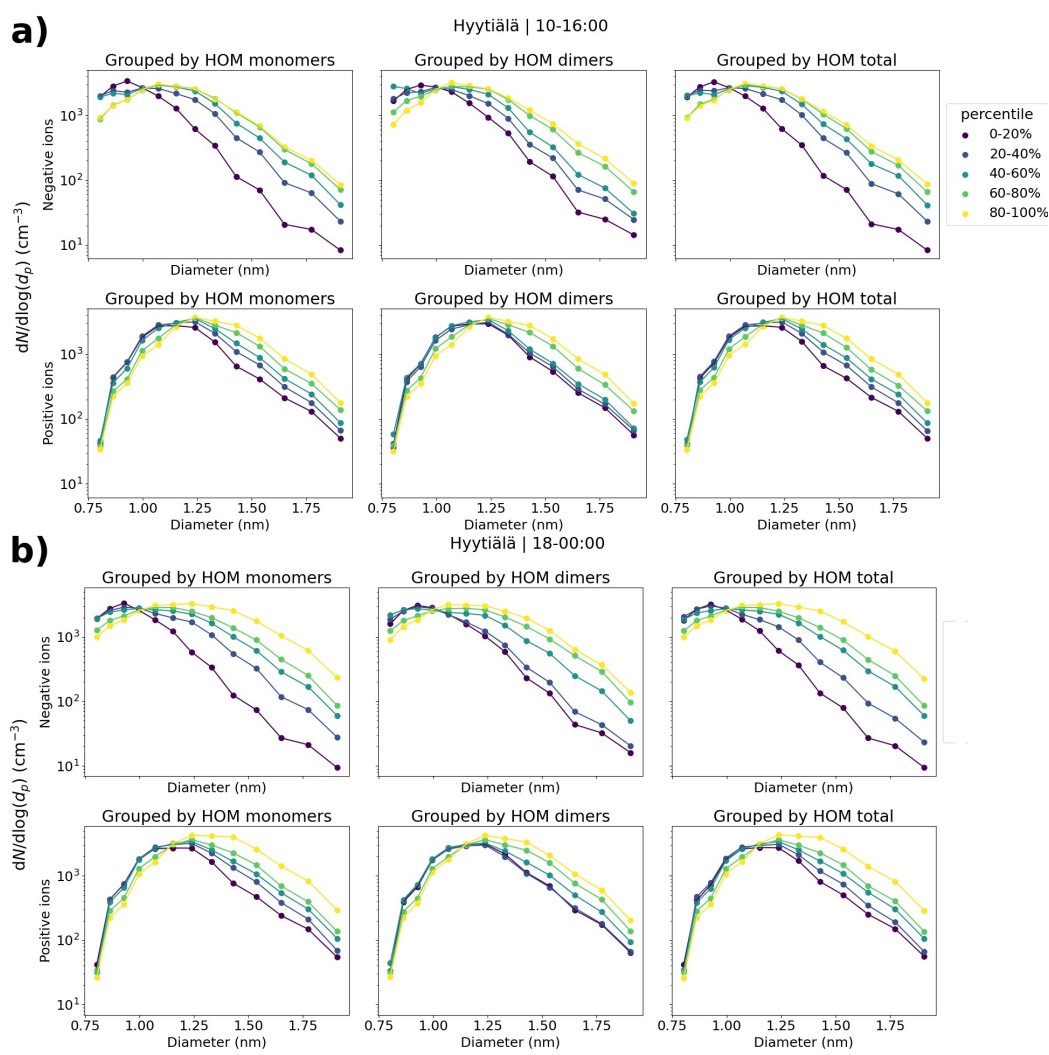

**Fig. 3:** The median negative and positive small ion (sub-2 nm) size distributions in Hyytiälä, Finland grouped by the percentiles of neutral HOM monomer, HOM dimer and total HOM concentrations. Both the evening (18:00-00:00) size distributions (b) and daytime (10:00-16:00) size distributions (a) are shown. Daytime percentiles for HOM monomers, dimers and total are 20%: $4.30 \cdot 10^6$, $5.93 \cdot 10^5$, and $5.10 \cdot 10^6$ cm$^{-3}$; 40%: $1.34 \cdot 10^7$, $1.45 \cdot 10^6$, and $1.50 \cdot 10^7$ cm$^{-3}$; 60%: $4.00 \cdot 10^7$, $2.44 \cdot 10^6$, and $4.18 \cdot 10^7$ cm$^{-3}$; 80%: $9.70 \cdot 10^7$, $7.62 \cdot 10^6$, and $1.05 \cdot 10^8$ cm$^{-3}$, respectively. Evening percentiles for HOM monomers, dimers and total are 20%: $3.00 \cdot 10^6$, $4.94 \cdot 10^5$, and $3.75 \cdot 10^6$ cm$^{-3}$; 40%: $8.40 \cdot 10^6$, $1.50 \cdot 10^6$, and $1.03 \cdot 10^7$ cm$^{-3}$; 60%: $3.17 \cdot 10^7$, $2.95 \cdot 10^6$, and $3.65 \cdot 10^7$ cm$^{-3}$; 80%: $7.52 \cdot 10^7$, $7.70 \cdot 10^6$, and $8.46 \cdot 10^7$ cm$^{-3}$, respectively.



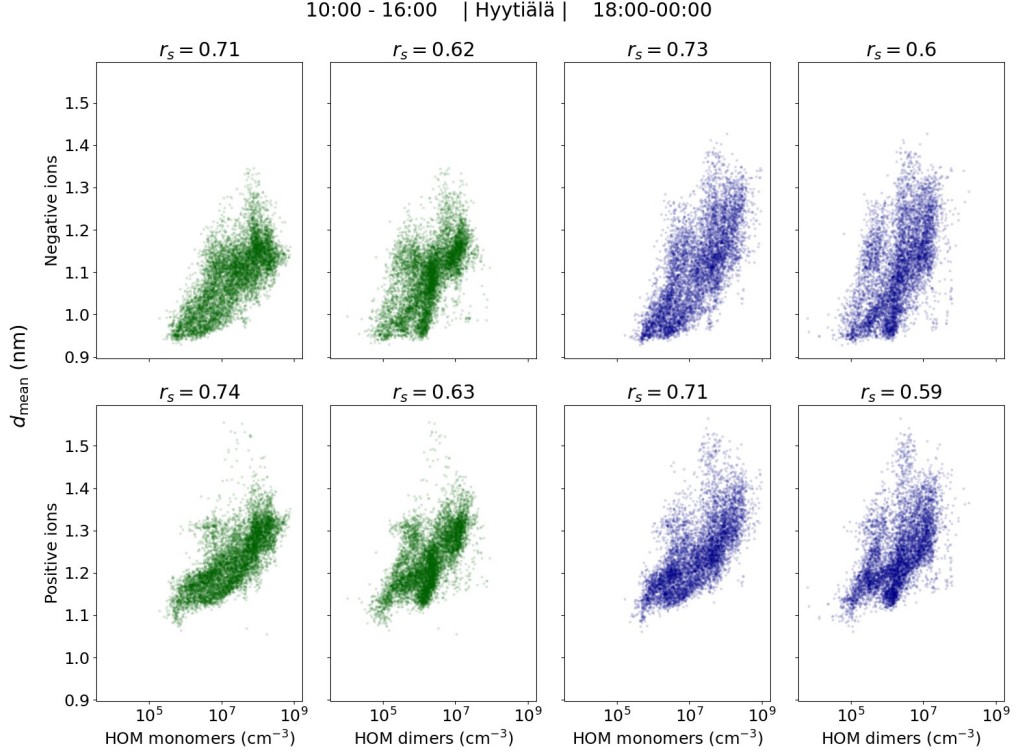

**Fig. 4:** Mean diameter ($d_{mean}$) of positive and negative small (sub-2 nm) ions as a function of HOM (monomer, dimer and total) concentration in Hyytiälä. The individual points are hourly medians, and the daytime (10:00-16:00, marked in green) and evening (18:00-00:00, marked in dark blue) are shown separately. Spearman correlation coefficients ($r_s$) are shown.

### 3.2.2 Sulfuric acid

Fig. 5 shows the median daytime negative and positive small ion (0.8-2 nm) size distributions grouped by percentiles of neutral sulfuric acid (SA) concentration, add percentiles here. In addition, the daytime hourly median $d_{mean}$ values are shown as a function of the SA concentration. We see a clear increase in the concentrations of negative (positive) small ions larger than approximately 1.05 (1.1) nm when comparing SA concentrations in the lower percentiles to the higher percentiles, until the behavior seems to stall so that the 60-80% and 80-100% percentiles show similar size distributions. From previous studies, we know that while sulfuric acid is often needed for the initial cluster formation, organic compounds tend drive cluster growth (Kulmala et al., 2013). This might partially explain the small difference in the size distributions between the 60-80% and 80-100% percentiles of the SA concentration.

A good positive correlation was seen between $d_{mean}$ and SA concentration for both polarities, $r_s$ = 0.51 (0.61) for negative (positive) ions (Fig. 5). The correlation is slightly weaker than was observed between $d_{mean}$ and HOM, especially monomer, concentrations. The majority of $d_{mean}$ values above 1.1 nm correspond to days with high NPF ranking, while most values of $d_{mean}$ below 1.1 nm correspond to days with low NPF rank values below 0.5. Notably, for $d_{mean}$ of negative small ions



above approx. 1.1 nm, the values of $d_{\mathrm{mean}}$ do not seem to increase with an increasing SA
concentration as clearly as they do with increasing HOM concentration (Fig. 4) . As discussed
above, organic compounds might be needed to drive the growth of small ions further, and thus
dependency of $d_{\mathrm{mean}}$ on SA is not seen as clearly when $d_{\mathrm{mean}}$ is above 1.1 nm.

Hyytiälä | 10:00 - 16:00

**Fig. 5:** The median number size distributions of small ions between 0.8 and 2 nm grouped by
percentiles of neutral sulfuric acid concentration (percentiles; top panel) and scatter plot and
Spearman correlation coefficients ($r_s$) of hourly mean diameter of small ions ($d_{\mathrm{mean}}$) and sulfuric acid
concentration (bottom panel) in Hyytiälä. In the scatter plot, the color indicates the respective NPF
rank of the day. Only daytime (10:00-16:00) values are included. The percentile values for sulfuric
acid are 20%: $4.22 \cdot 10^4$ cm$^{-3}$ , 40%: $1.79 \cdot 10^5$ cm$^{-3}$, 60%: $5.06 \cdot 10^5$ cm$^{-3}$ and 80%: $1.08 \cdot 10^6$ cm$^{-3}$.



## 3.3 Relationship of small ion size distribution with low volatility vapors in Beijing

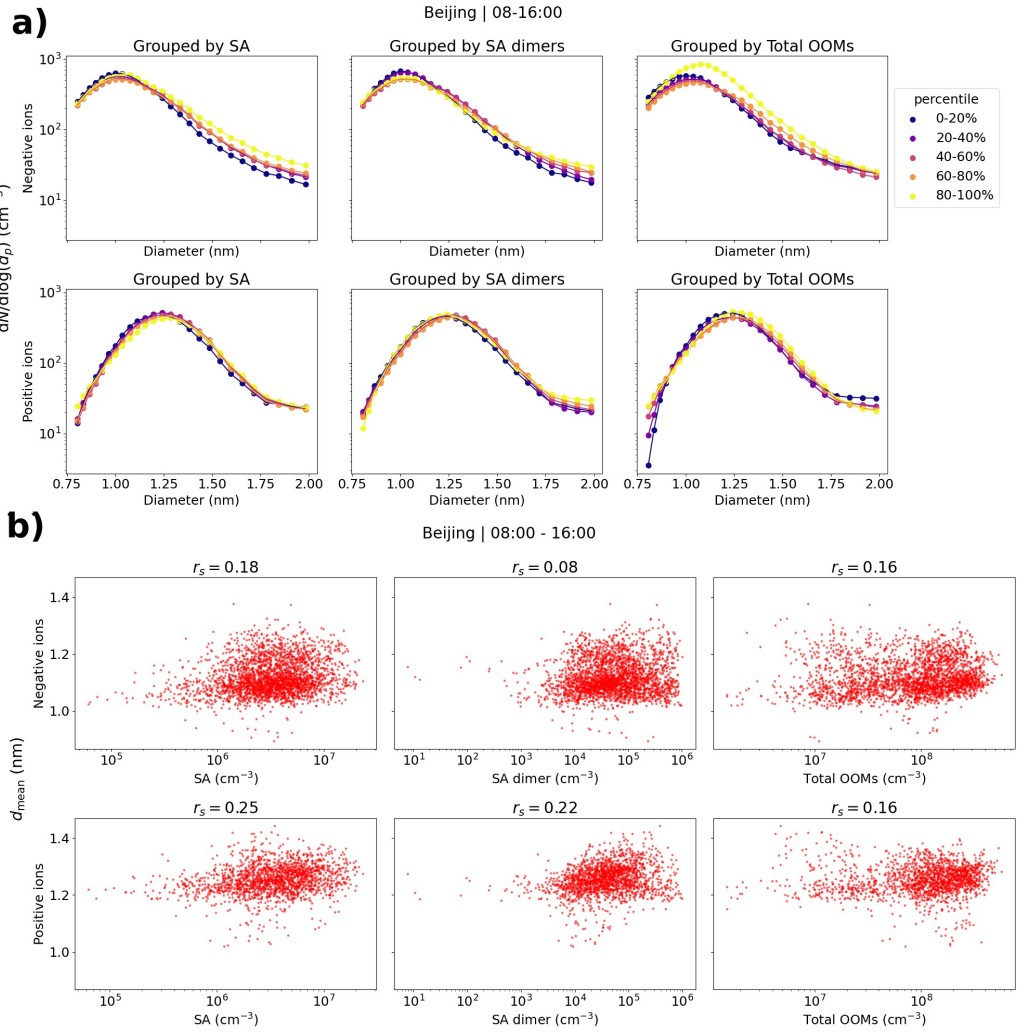

**Fig. 6:** (a) Small ion median daytime (08:00-16:00) number size distributions in Beijing, grouped by percentiles of sulfuric acid (SA), SA dimer or total oxidized organic molecule (OOM) concentrations. (b) Hourly daytime mean diameter ($d_{mean}$) of small ions versus SA, SA dimer and total OOM concentrations. Spearman correlation coefficients ($r_s$) are included. The percentile limits of SA, SA dimer and total OOM are 20%: $1.64 \cdot 10^6$, $1.26 \cdot 10^4$, and $2.19 \cdot 10^7$ cm⁻³; 40%: $2.53 \cdot 10^6$, $3.11 \cdot 10^4$, and $5.89 \cdot 10^7$ cm⁻³; 60%: $3.66 \cdot 10^6$, $6.01 \cdot 10^4$ and $1.23 \cdot 10^8$ cm⁻³; and 80%: $5.17 \cdot 10^6$, $1.32 \cdot 10^5$, and $2.15 \cdot 10^8$ cm⁻³.

Fig. 6a shows the number size distributions of small ions grouped by percentiles of neutral sulfuric acid, sulfuric acid dimer and total oxidized organic molecule (OOM) concentration in Beijing. We can see that, especially compared to results already for Hyytiälä, the differences in the size distributions are small for either polarity. The concentration of negative ions below approx. 1.2 nm





slightly decreases with increasing sulfuric acid and sulfuric acid dimer concentration, while the
concentrations above approx. 1.2 nm increase. Close to 2 nm, where the increase is the highest, the
concentration of ions is higher by around a factor of two when sulfuric acid concentration is in the
80-100% percentile compared to when it is in the 0-20% percentile. For both polarities, the
concentrations below approx. 1.75 nm appear higher when total OOM concentration is in the 80-
100% percentile compared to other times. However, the concentrations close 2 nm are not
simultaneously higher, indicating that despite the increased concentration of small ions, more of
them are not growing to intermediate ions.
Fig. 6b shows the scatter plots of $d_{mean}$ and sulfuric acid, sulfuric acid dimer and total OOM
concentrations. Weak positive correlation is seen, and the Spearman correlation coefficients ($r_s$) are
between 0.08 and 0.25. The differences in the values of $d_{mean}$ are small. The relationship between the
small ion size distribution or $d_{mean}$ and low volatility vapor concentrations in Beijing appears weak
and much less clear compared to Hyytiälä.  Due to the high concentration of both low volatility
vapors and large particles, the dynamics of small ions in a megacity such as Beijing are different
than in a rural site such as Hyytiälä.

## 3.4 Correlation of small ion size distribution with sulfuric acid clusters and NPF in Hyytiälä

Fig. 7a shows the median number size distribution of negative small ions grouped by percentiles of
the signals of SA ion clusters $HSO_4^-$ (monomer), $H_2SO_4 \cdot HSO_4^-$ (dimer) and $(H_2SO_4)_2 \cdot HSO_4^-$ (trimer)
and their ratios in Hyytiälä. The median distributions are determined from daytime (10:00-16:00)
values with clear sky conditions. We observe a clear increase in the number of small ions with
diameters above approx. 1.2 nm with the increased signal of SA ion monomers and dimers. sThe
increase is especially clear for dimers and the dimer to monomer ratio, and the concentration of
small ions close 2 nm, where the differences are highest, is an order of magnitude higher when
dimer signal is in the 80-100% percentile compared to when the signal is in the 0-20% percentile.
These results indicate that the dimer signal is a strong indicator for the cluster formation and the
growth of clusters to larger sizes in Hyytiälä.
Fig. 7b shows the median daytime (10:00-16:00) size distributions for both polarities with respect to
the percentiles of 2.0-2.3 nm ion concentrations and bins of NPF ranking (Aliaga et al., 2023) in
Hyytiälä. When the 2.0-2.3 nm ion concentration is higher, a clear increase in concentrations is seen
above approx. 1.2 nm. The difference in negative small ion concentrations close to 2 nm between
80-100% and 0-20% is over one order of magnitude. A high 2.0-2.3 nm ion concentration indicates
intense local-scale cluster formation and NPF (Tuovinen et al., 2024), and we can see from the
small ion size distribution for both polarities how this growth of small ions up to 2.0 nm is seen in
the small ion population as an increase in the concentrations of larger small ions.
Similar observations can be made from the small ion size distributions with respect to the different
NPF ranking values. However, the differences are smaller than with respect to 2.0-2.3 nm ions, and
especially for positive small ions such differences are very small. There is likely a combination of
factors at play here. First of all, NPF ranking was determined for total particles between 2.5 and 5
nm and there might be differences stemming both from the ranking being less sensitive for local
NPF and for 2.0-2.3 nm ion concentrations being more sensitive for ion-induced clustering or NPF.



In addition, differences between what is observed in the total particles versus ions can be caused by
variation in the chemical compounds, which take up the available charges (Bianchi et al., 2017).
We note that the differences in the number size distribution of positive small ions are once again
smaller than for negative small ions. Similarly to Sect. 3.2.1, we hypothesize that this is due to the
size difference between the polarities.
Fig. 8 shows the scatter plot of hourly daytime small ion $d_{\mathrm{mean}}$ and the concentration of 2.0-2.3 nm
ions. As expected, a strong positive trend is seen between $d_{\mathrm{mean}}$ and 2.0-2.3 nm ion concentrations.
The correlation coefficient is $r_s = 0.54$ (0.67) for negative (positive) ions. Fig. 8 also shows the box
plots of $d_{\mathrm{mean}}$ with NPF ranking. The median of $d_{\mathrm{mean}}$ increases with increasing NPF ranking, as
expected. However, the variance for lower rankings is much higher, resulting in overall quite a low
correlation between $d_{\mathrm{mean}}$ and NPF ranking, $r_s =$ for negative (positive small) ions.

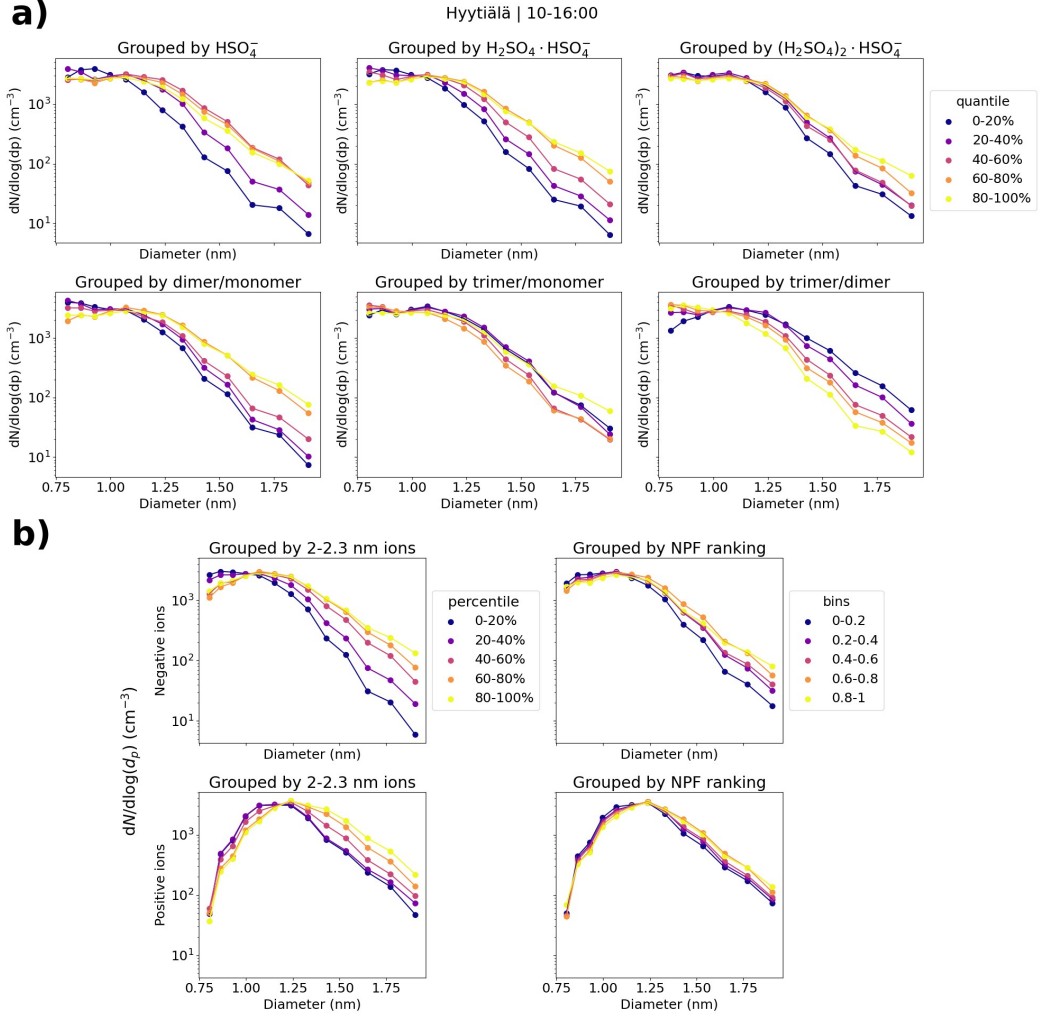




**Fig. 7:** (a) Hyytiälä daytime median negative small ion number size distributions grouped by percentile of the signals of $HSO_4^-$, $H_2SO_4\cdot HSO_4^-$ or $(H_2SO_4)_2\cdot HSO_4^-$ ions and their ratios. (b) Daytime median small ion size distributions for both polarities grouped by the percentile of 2.0-2.3 nm ion concentrations of the respective polarity or by NPF ranking. The percentile limits for negative (positive) 2.0-2.3 nm ion concentrations are 20%: 0.48 (1.75) cm$^{-3}$, 40%: 1.01 (2.34) cm$^{-3}$, 60%: 1.94 (3.49) cm$^{-3}$, and 80%: 3.39 (5.35) cm$^{-3}$.

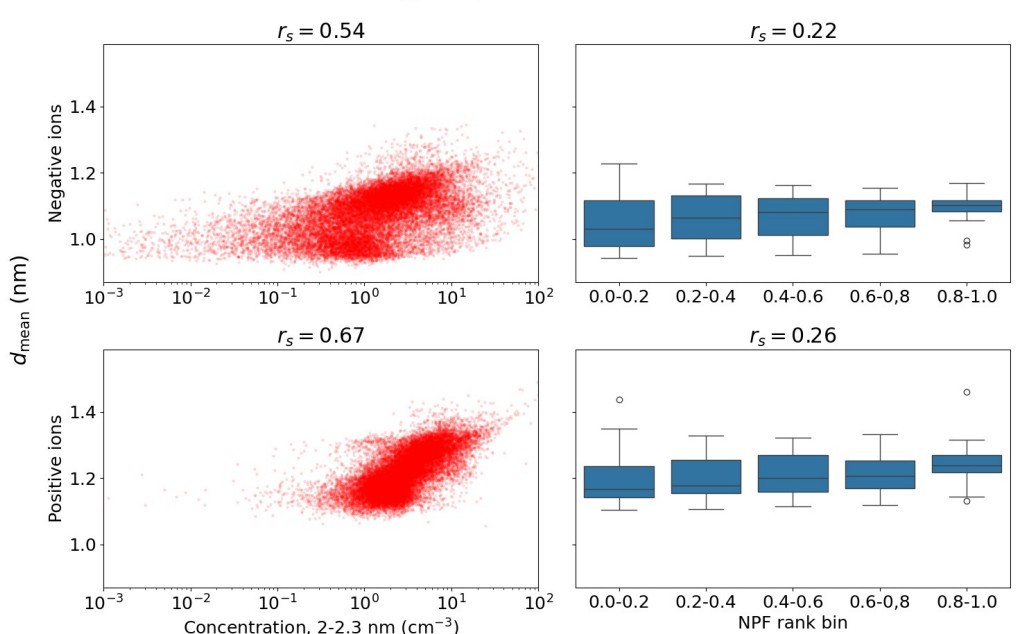

**Fig. 8:** Hourly daytime negative and positive small ion diameter versus concentration of 2.0-2.3 nm ions of respective polarity or NPF ranking in Hyytiälä. Correlation coefficients ($r_s$) are also shown. The middle line of the box plots for $d_{mean}$ and NPF rank are the median values, while the boxes show the 25% and 75% percentiles and the lines the 10% and 90% percentiles.



## 3.5 Impact of NPF on small ion distribution in Beijing

**a)**

**b)**

**Fig. 9:** (a) Median negative and positive small ion number size distributions in Beijing grouped by
percentiles of 2.0-2.3 nm ion concentrations (of respective polarity) or by NPF ranking. (b). Scatter





plots of mean diameter and 2.0-2.3 nm ion concentrations (of respective polarity) or NPF ranking.
Values are for daytime (08:00-16:00). The percentile limits of 2.0-2.3 nm concentration for negative
(positive) ions are 20%: 0.51 (0.85) cm$^{-3}$, 40%: 0.92 (1.27) cm$^{-3}$, 60%: 1.33 (1.72) cm$^{-3}$, and 80%:
2.47 (3.22) cm$^{-3}$.
Fig. 9a shows the small ion size distributions with respect to the concentration of 2.0-2.3 nm ions or
NPF ranking in Beijing. For both polarities, clear differences are seen in the distributions depending
on the percentile of the 2.0-2.3 nm ion concentration. When the 2.0-2.3 nm ion concentrations are
higher, the concentration of negative (positive) small ions above approx. 1.0 (1.3) nm is increased.
Relative to the concentrations, the differences are largest close to 2 nm. Comparing the 0-20% and
80-100% percentiles, the difference in concentrations is around one order of magnitude when the
diameter is approaching 2 nm. Similar observations are seen with respect to NPF ranking, although
to a lesser extent. For negative small ions, the concentration at around 2 nm is four to five times
higher when the NPF ranking is above 0.80 compared to when it is below 0.20. For positive ions,
the concentration is less than two times higher.
When looking at the small ion distributions in Beijing for different 2.0-2.3 nm ion concentrations or
NPF ranking, unlike for low-volatility vapor concentrations, we are able to see the impact of growth
of small ions to intermediate ions in the size distribution. In Beijing, CoagS is crucial in
determining whether the growing clusters will survive to larger sizes or not, and therefore, even if
the concentrations of precursors are high, growth might be negligible. A high 2.0-2.3 nm ion
concentration or NPF rank means that a considerable number of clusters are able to grow without
being scavenged by pre-existing larger particles.
Fig. 9b shows the scatter plots of $d_{mean}$ and 2.0-2.3 nm ion concentration and the box plots of $d_{mean}$
and NPF ranking. The correlation coefficients for negative ions are as expected, $r_s$ =0.58 and $r_s$ =
0.41 between $d_{mean}$ and 2.0-2.3 nm ion concentration or NPF ranking, respectively. For positive ions,
the correlation coefficient between $d_{mean}$ and NPF ranking is $r_s$ = 0.59, while it is only 0.21 between
$d_{mean}$ and 2.0-2.3 nm ion concentration. From Fig. 10a we see that the concentrations of positive
small ions below 1 nm also increase to some extent with increasing 2.0-2.3 nm ion concentration,
which likely impacts the values of $d_{mean}$, resulting in a relatively poor overall correlation.
Notably, the differences in size distributions with respect to NPF ranking are clearer and the
correlation between $d_{mean}$ and ranking is stronger in Beijing than in Hyytiälä for both polarities. The
explaining factors could be the fact that intense NPF in Beijing is more common than in Hyytiälä
(e.g., Dada et al., 2017; Deng et al., 2020), impacting the statistics of the ranking, and that local
clustering events, where ions or particles grow close to 2 nm but not much further, could be more
common in Hyytiälä. Overall, our results show that compared to a rural boreal forest site like
Hyytiälä, the dynamics of sub-2 nm ions in a polluted megacity like Beijing are different.

## 443 3.6 Case studies

Next, some case studies into the development of negative small ion size distributions, and other
investigated variables, are presented for Hyytiälä and Beijing. These cases show that we are able to
observe the cluster growth, driven by daytime NPF or evening clustering, from the ion number size
distributions of individual days and not only from the statistics of the size distributions. Based on
the analysis presented in this study, the behavior of negative and positive small ion populations is
mostly similar, and therefore, for simplicity, we have limited the analysis here to negative polarity.



### 3.6.1 Hyytiälä case 1 – an early spring day with NPF

First of the investigated days was 10[th] of March, in 2021 and is presented in Fig. 10. During this day, a strong NPF event was observed with clear growth observed both in the total particle and ion number size distribution (see Fig. A1). In the morning, a strong increase in the SA ion dimer and trimer signals was detected after 07:00 (Fig. 10c), which occurred simultaneously with an increase in the concentration of neutral sulfuric acid. Shortly after, at around 08:00, neutral HOM monomer concentration started to increase (Fig. 10d). A strong increase in the concentration of 2.0-2.3 nm negative ions was observed after 09:00, indicating intense NPF on a local-scale (Fig. 10a). Approximately one hour before an increase in the concentration of 2.0-2.3 nm ions was first observed, the small ion $d_{mean}$ started to increase from below 1.0 nm (Fig. 11a), showing that growth of clusters in the small ion population to larger sizes had begun. We can also see this from the negative ion number size distributions (0.8-2 nm; Fig. 10b): in the early hours of the day, the concentrations of the smallest ions are at their highest while the concentration of ions above approx. 1.1 nm are at their lowest. Throughout the morning hours, we can see that the concentration of ions above approx. 1.1 nm increases and in the afternoon, around 14:00, the concentration of ions close to 2 nm is over a order of magnitude higher than during the night before. At around 14:00, the concentration of 2.0-2.3 nm ions and small ion $d_{mean}$ also reach their peaks. Then, the concentrations of larger small ions, 2.0-2.3 nm and SA ion clusters starts to decrease, alongside with the concentration of HOM monomers.

We also took a look at the diameter specific concentrations in a smaller time frame (Fig. A4), which clearly shows how clear increase concentrations is observed for the diameters above 1.2 nm. A time delay between the increasing concentration of larger ions and smaller ions was seen, showing the growth of ions between 1.2 to 2 nm. Using the appearance time method (Lehtipalo et al., 2014), GR between 1.24 to 2.05 nm was estimated: GR = 0.40 nm/h. This value is somewhat lower than typical GRs reported in Hyytiälä (Hirsikko et al., 2005; Yli-Juuti et al., 2011), as expected due to the small size of the considered ions. Regardless, it shows that the growth of ions below 2 nm is non-negligible.

This day clearly shows the connection between sulfuric acid and HOMs with particle formation, and illustrates how the small ion number size distribution changes as the ions grow from close to 1 nm in diameter to above 2 nm.





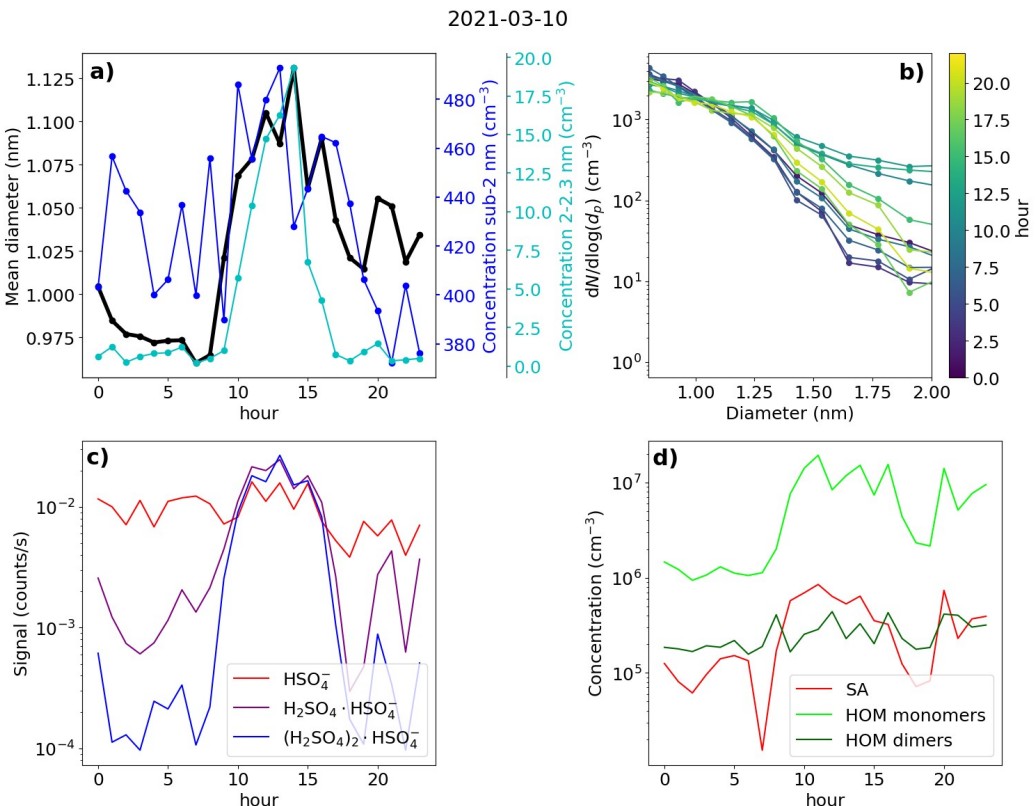

**Fig. 10:** Data from Hyytiälä, 10th of March, 2021. (a) Hourly mean diameter of negative small ions (0.8-2.0 nm), total concentration of small ions, and concentration of 2.0-2.3 nm negative ions. (b) Two-hour median number size distribution of negative small ions. (c) Hourly signals of $HSO_4^-$, $H_2SO_4 \cdot HSO_4^-$ and $(H_2SO_4)_2 \cdot HSO_4^-$ ions. (d) Hourly median concentrations of neutral sulfuric acid (SA) and highly oxidized molecule (HOM) monomers and dimers.

### 3.6.2 Hyytiälä Case 2 – a spring day with strong evening clustering

The second of the chosen days for Hyytiälä is 19th of April, 2021 (Fig. 11). The NPF ranking of this day was high, over 0.9, however the growth in the negative ion and total particle mode was discontinuous with the clearest growth observed above 5 nm, suggesting that the fraction of growing locally formed neutral clusters or ions was low (Fig. A2). However, strong evening ion cluster formation was observed on this day. Therefore, Case 2 illustrates both the probable contribution of organic vapors to initiate the growth of larger particles and the evening ion cluster formation attributable to HOM dimers (Mazon et al., 2016).

Starting from the early hours of the day, the signals of SA ions and neutral SA concentration increase (Fig. 11c and 11d), reaching their maxima around 13:00 in the early afternoon. Compared to Hyytiälä Case 1, the signal from trimers is lower in relation to the signal from monomer and dimer. From the negative ion number size distributions (Fig. 11b), we see that the concentration of negative ions below approx. 1.2 nm increases and the concentration of small ions above approx. 1.2 nm strongly decreases starting from the early hours of the day until afternoon. This is reflected in



the value of $d_{mean}$, which decreases from over 1.2 nm to below 1.1 nm (Fig. 11a). The concentration
of 2.0-2.3 nm negative ions decreases until 08:00 in the morning, after which it increases briefly
before decreasing again (Fig. 11a). The small ion total concentration also strongly decreases from
over 800 cm$^{-3}$ to 600 cm$^{-3}$ (Fig. 11a). Unlike in Case 1, on this day, the growth of small ions during
daytime is negligible and an increased fraction of the available charge is taken up by small, below
1.2 nm ions, many of which are likely composed of sulfuric acid monomers or dimers. This
explains the behavior of the ion size distributions, $d_{mean}$ and the total small ion concentration.
After 14:00 in the afternoon, the concentration of neutral HOM dimers starts to increase, and
reaches a peak at around 19:00 (Fig. 11d). Compared to Case 1, the HOM dimer concentration is
over one order of magnitude higher. Notably, at the same time as the HOM dimer concentration
starts increase, clear growth of total particles above 5 nm is observed (Fig. A2). Concentration of
small ions larger than approx. 1.2 nm (Fig. 11b) and 2.0-2.3 nm ion concentration (Fig. 11a)
strongly increase. Small ion $d_{mean}$ increases from approx. 1.1 nm to 1.3 nm, while the total negative
small ion concentration increases from around 600 cm$^{-3}$ to 1000 cm$^{-3}$. The negative ion GR between
1.43 to 2.05 nm was estimated to be 1.28 nm/h (Fig. A4), which is over twice as high as the GR
estimated for Case 1, likely due to the high concentration of lower volatility HOMs driving the
small ion growth.

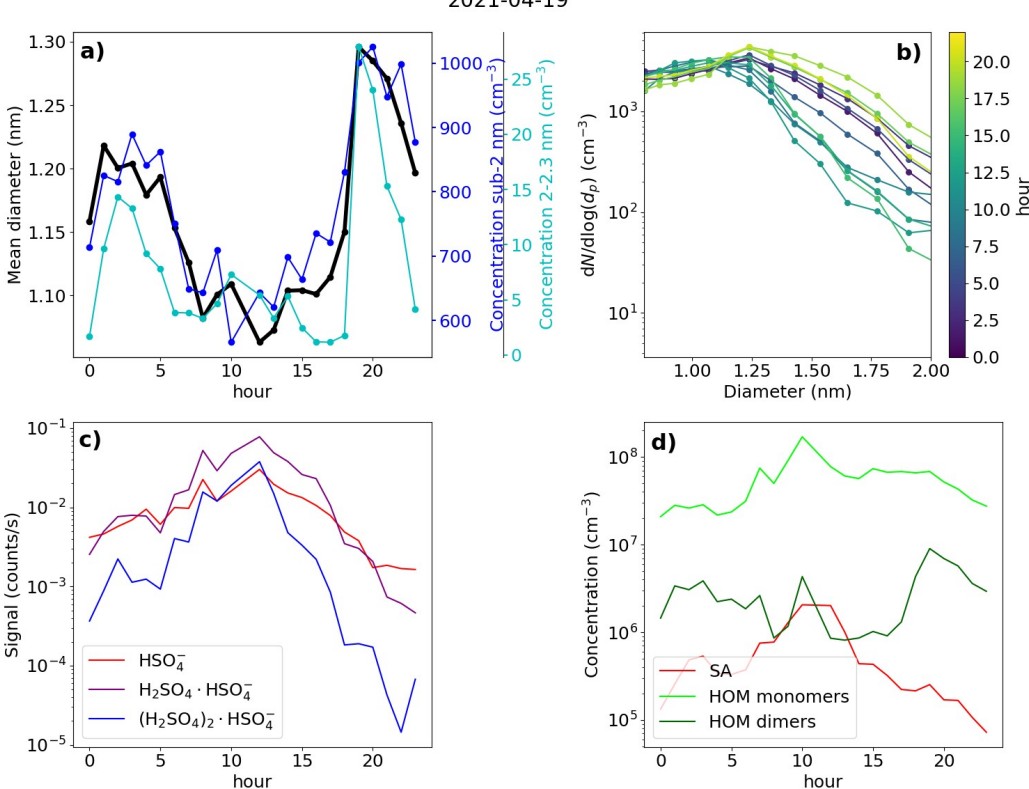

**Fig. 11:** Data from Hyytiälä, 19$^{th}$ of April, 2021. (a) Hourly mean diameter of negative small ions
(0.8-2.0 nm), total concentration of small ions, and concentration of 2.0-2.3 nm negative ions. (b)

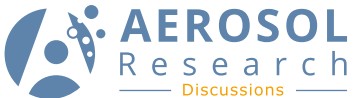

Two-hour median number size distribution of negative small ions. (c) Hourly signals of $HSO_4^-$,
$H_2SO_4 \cdot HSO_4^-$ and $(H_2SO_4)_2 \cdot HSO_4^-$ ions. (d) Hourly median concentrations of neutral sulfuric acid
(SA) and highly oxidized molecule (HOM) monomers and dimers.

***3.6.3 Beijing Case 1 – a day with intense NPF***



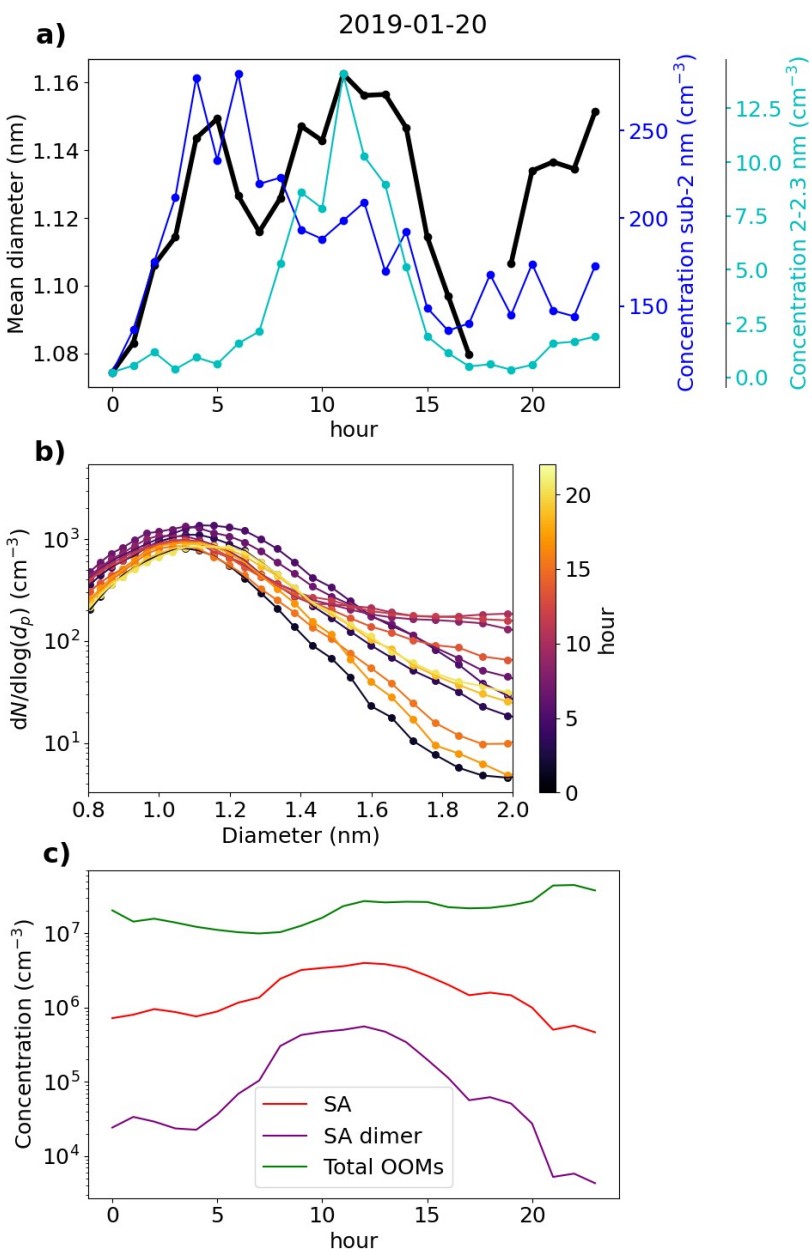

**Fig. 12:** Data from Beijing, 20th of January, 2019. (a) Hourly mean diameter of negative small ions (0.8-2.0 nm), total concentration of small ions, and concentration of 2.0-2.3 nm negative ions. (b) Two-hour median number size distribution of negative small ions. (c) Hourly median concentrations of neutral sulfuric acid (SA), SA dimer and total oxidized organic molecules (OOMs).



Fig. 12 presents data from Beijing on 20[th] of January, 2019. This day was characterized by an intense NPF event, observed both in the ion and total particle size distribution (Fig. A3). We see that from 00:00 until 05:00 in the morning, the negative small ion concentrations seem to increase, which is apparent for the whole sub-2 nm size range (Fig. 12a and 12b). The concentration of 2.0-2.3 nm negative ions stays low (Fig. 12a), indicating that there is no significant growth of small ions to intermediate ions. The increase in small ion concentration could be due to reduction in CS or even meteorological conditions.

After 05:00 in the morning, an increase in neutral sulfuric and sulfuric acid dimer concentration is observed (Fig. 12c). Simultaneously, the concentration of 2.0-2.3 nm ions increases sharply, indicating the formation of intermediate ions. Two changes in the small ion size distribution are shown: first, the concentration of small ions below approx. 1.5 nm ions, decreases and second, the concentration of small ions above that increases. Increasing growth of small ions to larger sizes causes a shift in their size distribution. Notably, no growth in the surface plots (Fig. A3) is observed yet, likely due to locality of or insufficient intensity of the ion formation. After 12:00, the concentrations of small ions larger than approx. 1.5 nm start to decrease, as does the concentration of 2.0-2.3 nm ions.  While the growth of ions and particles at larger diameters continues, the intensity of cluster growth decreases.

In this case, the negative small ion GR was estimated to be 0.24 nm/h from 1.72 to 2.06 nm (Fig. A5), which is lower than the values determined for the two Hyytiälä cases and is on the lower range of values of particle GRs for Beijing (Deng et al., 2020). Another noteworthy observation can be made from the diameter specific concentrations (Fig. A5): as already seen from the size distributions and more clearly here, the concentrations of ions up to around 1.5 nm decrease, while the concentrations above increase at the same time. This implies that the ions, which actually start to grow to larger sizes are close to 1.5 nm in diameter, though at such a low GR their survival probability to larger sizes is likely very low (Kulmala et al., 2017).

In Sect. 3.3, we saw how in Beijing there appeared to be no correlation between the small ion number size distribution and the concentration of sulfuric acid. However, this day shows that despite the poor overall correlation, on some days there does appear to simultaneously be an increase in sulfuric acid concentration, and an increase in the growth of small ions.

## 4 Conclusions

We studied the seasonality of small ion number size distribution and the relationship of the small ion size distribution with low-volatility organic vapors, sulfuric acid and NPF in a rural boreal forest location of Hyytiälä, Finland and an urban megacity location of Beijing, China. Both analysis of long time series of data and daily case studies were carried out. We found a clear seasonality of the small ion size distribution in Hyytiälä, where the small ions of both polarities were the smallest in size during winter and the largest during late spring and summer. In Beijing, while there were month-to-month variations in the size distribution, but no clear seasonal pattern was identified, which we note could partly be due to the smaller number of data from Beijing compared to Hyytiälä.

We found that in Hyytiälä the small ion size distribution strongly varied with respect to the concentration of organic, especially highly oxidized organic (HOM) monomer, compounds and that



the concentration of small ions above approx. 1.2 nm increased strongly with increasing HOM monomer concentration. This was observed more strongly for negative polarity and during the evening, which was found to be connected to the evening ion cluster formation driven by organics in Hyytiälä. The small ion size distribution also showed clear increase in the size of the small ions in Hyytiälä with respect to neutral sulfuric acid and ionized sulfuric acid dimers, associated with daytime cluster formation and growth. In contrast, there was no clear relationship between the concentration of either organic vapor or sulfuric acid and the size of the small ions in Beijing. The reason for this remains to be unidentified, but we hypothesize that the high scavenging loss rate of ions could suppress or hide the impact of these vapors on the small ion size distribution.

When the concentration of ions in the range 2.0-2.3 nm increased, indicating the occurrence of local NPF, we observed clear signs of growth in the small ion size distribution. This was seen in both locations, even in Beijing, where no clear association of small ion size with organic vapor or sulfuric acid was found. To a lesser extent, an increase in the small ion size was also seen with respect to NPF rank, a parameter, which characterizes the intensity of NPF.

Overall, our results have shown in a novel way how the atmospheric cluster formation and growth processes impact the number size distribution of small ions. We have also shown how the small ion size distribution can be used to observe and get insight into these processes.

## Author contributions

ST analyzed the data and wrote the manuscript. JL was responsible for the ion measurements in Hyytiälä. CL and NS were responsible for the measurements of low-volatility vapors and ion clusters. YL was responsible for the measurements in Beijing. MK and VMK conceptualized the study. All authors contributed to reviewing and editing the manuscript.

## Code and data availability

The data and the codes used in the analysis and to produce the figures are available upon request from the authors.

## Competing interests

At least one of the (co-)authors is a member of the editorial board of Aerosol Research. Authors have no other competing interests to declare.

## Acknowledgments

This work has been supported by the ACCC Flagship funded by the Academy of Finland grant nos. 337549 (UH) and 337552 (FMI), and the "Gigacity" project funded by the Jenny and Antti Wihuri Foundation. We acknowledge the SMEAR II and AHL/BUCT technical and scientific staff.



607 **Appendix**

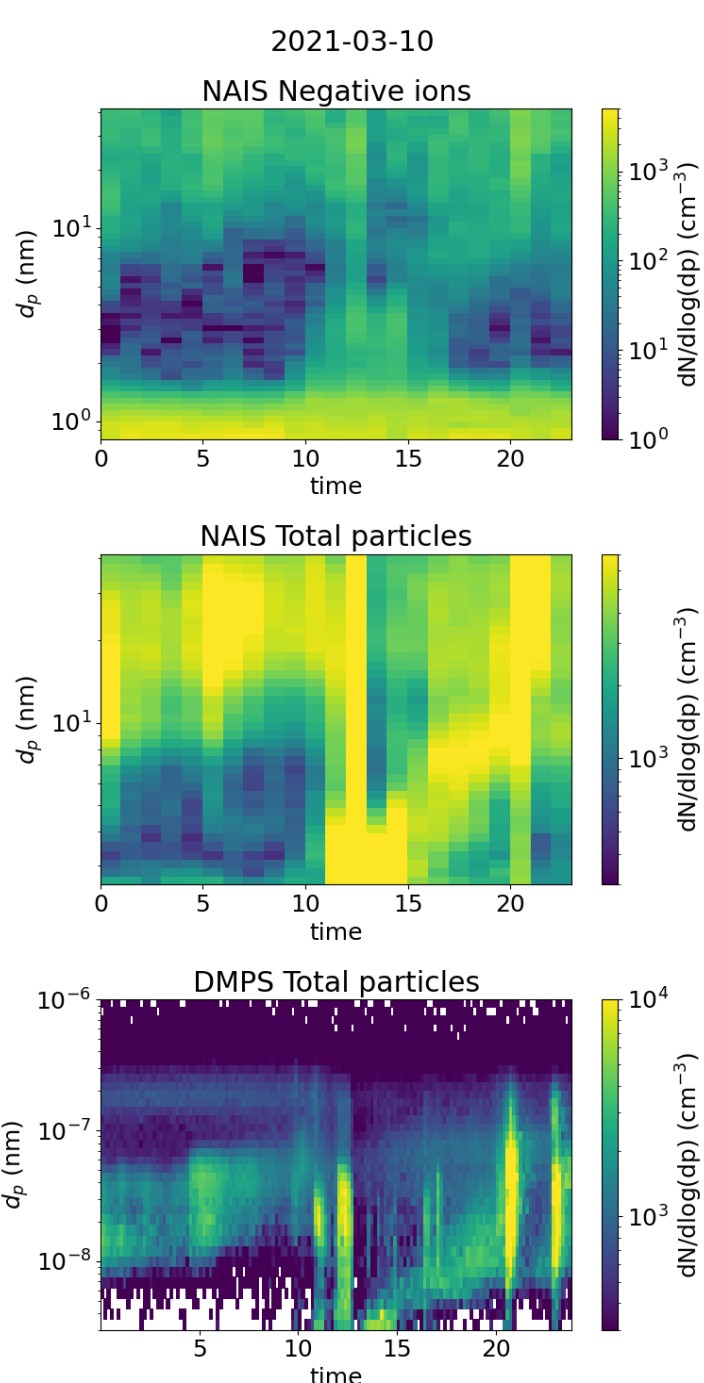

608





612 **Fig. A2:** Surface plots of negative ion number size distribution and total particle number size
613 distribution measured by NAIS and DMPS in Hyytiälä on 19th of April, 2021.

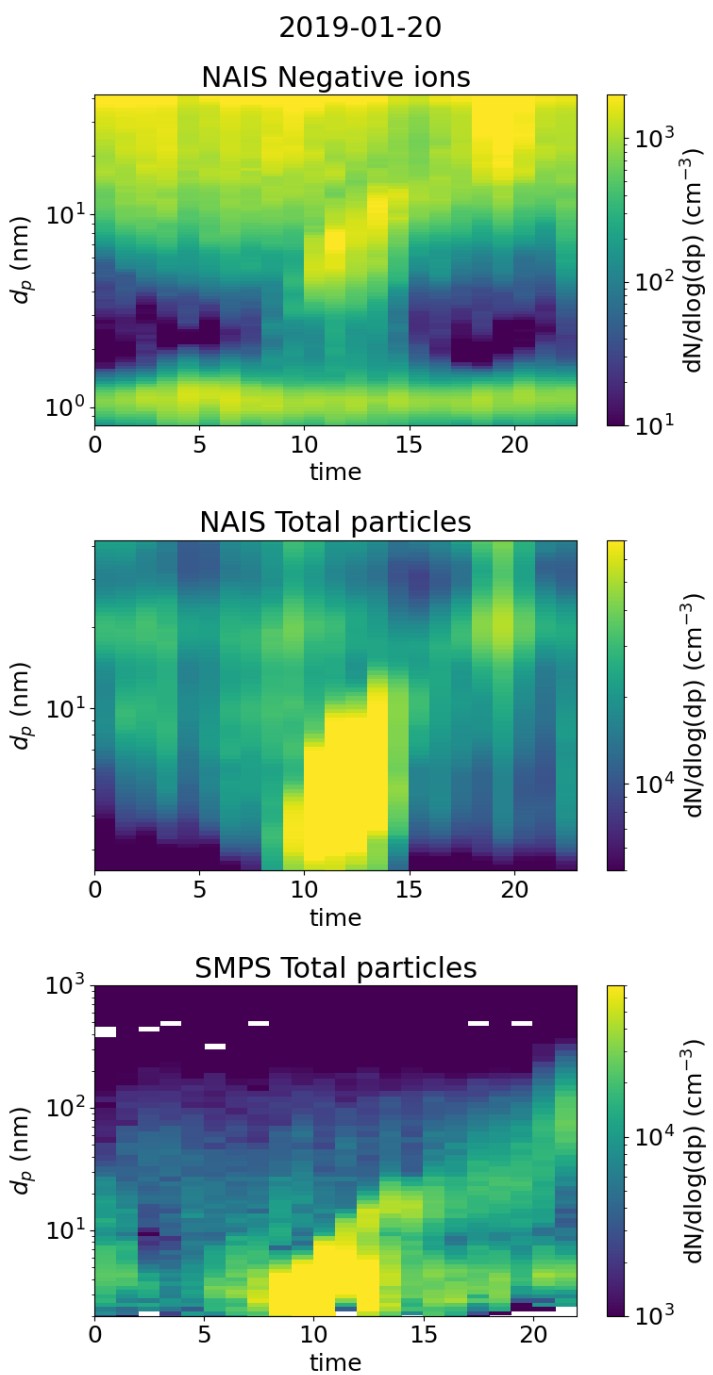

614





**Fig. A3:** Surface plots of negative ion number size distribution and total particle number size distribution measured by NAIS and SMPS (see Liu et al., 2016 for more information) in Beijing on 20[th] of January, 2019.



**Fig. A4:** The upper panels show concentrations of ions of a certain diameter with the hour of the day on 10[th] of March, 2021 and 19[th] or April, 2021 in Hyytiälä, Finland. The different colors of the line indicate the respective ion diameter ($d_i$). The bottom panels show the appearance time, defined as the time that the concentration reaches 50% of its maximum, and the respective $d_i$. The ion growth rate (GR) derived from these values as a slope of linear regression is shown. For 10[th] of March, the GR was determined from 1.24 to 2.05 nm and for 19[th] of April from 1.43 to 2.05 nm.



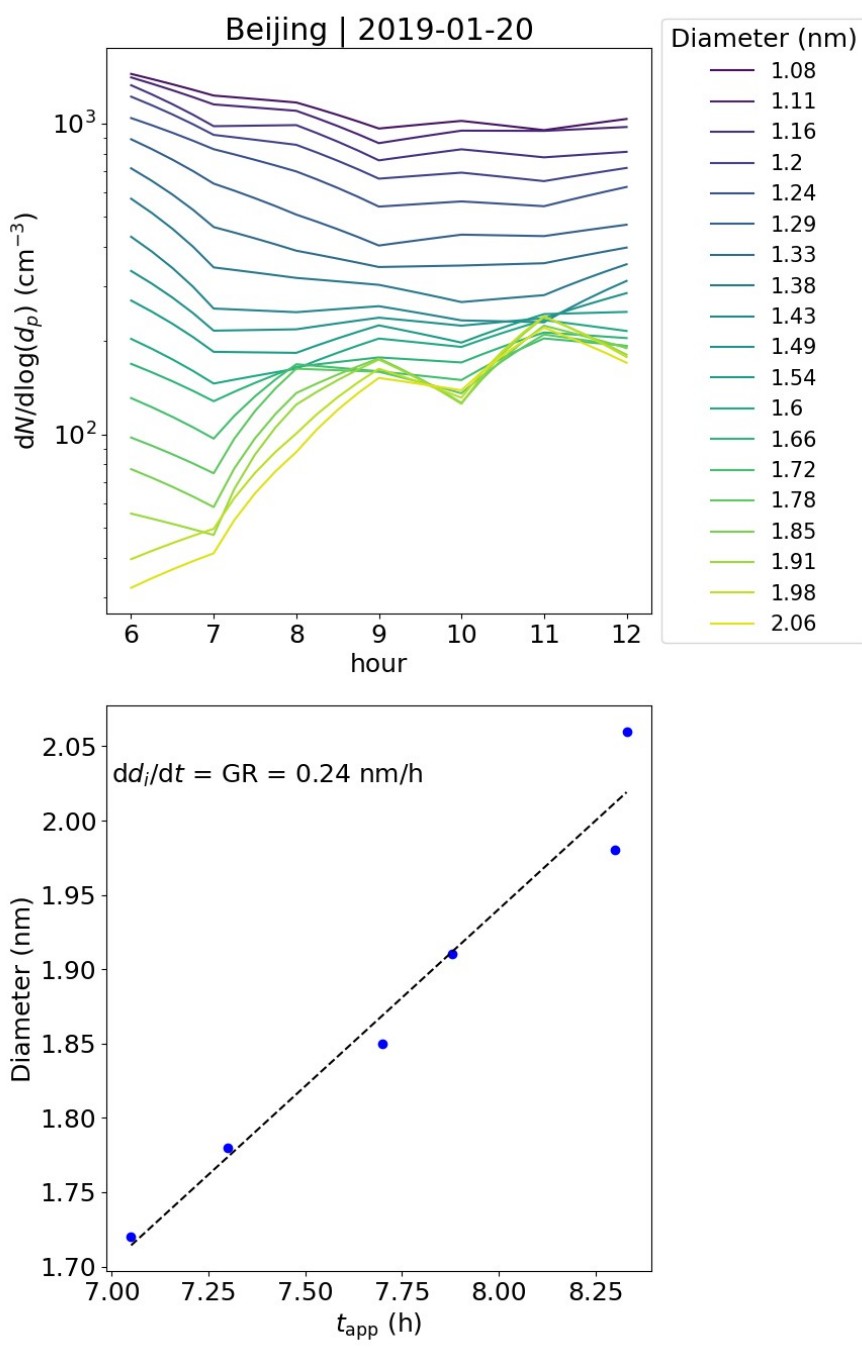

**Fig. A5:** The upper panel shows the concentrations of ions of a certain diameter with the hour of the day on 20$^{th}$ of January, 2019 Beijing, China. The different colors of the line indicate the respective



ion diameter ($d_i$). The bottom panel shows the appearance time, defined as the time that the
concentration reaches 50% of its maximum, and the respective $d_i$. The ion growth rate (GR)
derived from these values as a slope of linear regression is shown. The GR was determined from
1.72 to 2.06 nm.

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

Methods for determining particle size distribution and growth rates between 1 and 3 nm using the
Particle Size Magnifier, Boreal Environ. Res., 19, 215–236, 2014.

Lehtipalo, K., Yan, C., Dada, L., Bianchi, F., Xiao, M., Wagner, R., Stolzenburg, D., Ahonen, L. R.,
Amorim, A., Baccarini, A., Bauer, P. S., Baumgartner, B., Bergen, A., Bernhammer, A.-K.,
Breitenlechner, M., Brilke, S., Buckholz, A., Mazon, S. B., Chen, D., Chen, X., Dias, A., Dommen,
J., Draper, D. C., Duplissy, J., Ehn, M., Finkenzeller, H., Fisher, L., Frege, C., Fuchs, C., Garmash,
O., Gordon, H., Hakala, J., He, X. C., Heikkinen, L., Heinrizi, M., Helm, J. C., Hofbauer, V., Hoyle,
C. R., Jokinen, T., Kangasluoma, J., Kerminen, V.-M., Kim, C., Kirkby, J., Kontkanen, J., Kürten,
A., Lawler, M. J., Mai1, H., Mathot, S., Mauldin III, R. L., Molteni, U., Nichman, L., Nie, W.,
Nieminen, T., Ojdanic, A., Onnela1, A., Passananti, M., Petäjä, T., Piel, F., Pospisilova, V.,
Quéléver, L. L. J., Rissanen, M. P., Rose, C., Sarnela, N., Schallhart, S., Sengupta, K., Simon, M.,
Tauber, C., Tomé, A., Tröst, J., Väisänen, O., Voge, A. L., Volkamer, R., Wagner, A. C., Wang, M.,
Weitz, L., Wimmer, D., Ye, P., Ylisirniö, A., Zha, Q., Carslaw, K., Curtius, J., Donahue, N., Flagan,
R. C., Hansel, A., Riipinen, I., Virtanen, A., Winkler, P. M., Baltensperger, U., Kulmala, M., and
Worsnop, D. R.: Multi-component new particle formation from sulfuric acid, ammonia and biogenic
vapors, Sci. Adv., 4, eaau5363, https://doi.org/10.1126/sciadv.aau5363, 2018.

Li, J., Carlson, B. E., Yung, Y. L., Lv, D., Hansen, J., Penner, J. E., Liao, H., Ramaswamy, V., Kahn,
R. A., Zhang, P., Dubovik, O., Ding, A., Lacis, A. A., Zhang, L., and Dong, Y.: Scattering and
absorbing aerosols in the climate system, Nature Reviews Earth & Environment, 3, 363–379,
https://doi.org/10.1038/s43017-022-00296-7, 2022.

Liu, J. Q., Jiang, J. K., Zhang, Q., Deng, J. G., and Hao, J. M.: A spectrometer for measuring
particle size distributions in the range of 3 nm to 10 μm, Front. Env. Sci. Eng., 10, 63–72,
https://doi.org/10.1007/s11783-014-0754-x, 2016.





Liu, Y., Yan, C., Feng, Z., Zheng, F., Fan, X., Zhang, Y., Li, C., Zhou, Y., Lin, Z., Guo, Y., Zhang,
Y., Ma, L., Zhou, W., Liu, Z., Dada, L., Dällenbach, K., Kontkanen, J., Cai, R., Chan, T., Chu, B.,
Du, W., Yao, L., Wang, Y., Cai, J., Kangasluoma, J., Kokkonen, T., Kujansuu, J., Rusanen, A., Deng,
C., Fu, Y., Yin, R., Li, X., Lu, Y., Liu, Y., Lian, C., Yang, D., Wang, W., Ge, M., Wang, Y., Worsnop,
D. R., Junninen, H., He, H., Kerminen, V.-M., Zheng, J., Wang, L., Jiang, J., Petäjä, T., Bianchi, F.,
and Kulmala, M.: Continuous and comprehensive atmospheric observations in Beijing: a station to
understand the complex urban atmospheric environment, Big Earth Data, 4, 295–321,
https://doi.org/10.1080/20964471.2020.1798707, 2020
Mirme, S. and Mirme, A.: The mathematical principles and design of the NAIS – a spectrometer for
the measurement of cluster ion and nanometer aerosol size distributions, Atmos. Meas. Tech., 6,
1061–1071, https://doi.org/10.5194/amt-6-1061-2013, 2013.
Mirme, S., Balbaaki, R., Manninen, H. E., Koemets, P., Sommer, E., Rörup, B., Wu, Y., Almeida, J.,
Ehrhart, S., Weber, S. K., Pfeifer, J., Kangasluoma, J., Kulmala, M., and Kirkby, J.: Design and
performance of the Cluster Ion Counter (CIC), Atmos. Meas. Tech. Discuss. [preprint],
https://doi.org/10.5194/amt-2024-138, accepted for publication, 2024.
Quaas, J., Ming, Y., Menon, S., Takemura, T., Wang, M., Penner, J.E., Gettelman, A., Lohmann, U.,
Bellouin, N., Boucher, O., Sayer, A.M., Thomas, G.E., McComiskey, A., Feingold, G., Hoose, C.,
Kristjánsson, J.E., Liu, X., Balkanski, Y., Donner, L. J., Ginoux, P.A., Stier, P., Grandey, B.,
Feichter, J., Sednev, I., Bauer, S.E., Koch, D., Grainger, R.G., Kirkev˚ag, A., Iversen, T., Seland, Ø.,
Easter, R., Ghan, S.J., Rasch, P.J., Morrison, H., Lamarque, J.-F., Iacono, M. J., Kinne, S., and
Schulz, M.:Aerosol indirect effects – general circulation mode intercomparison and evaluation with
satellite data, Atmos. Chem. Phys., 9, 8697–8717, doi:10.5194/acp-9-8697-2009, 2009.
Rose, C., Zha, Q., Dada, L., Yan, C., Lehtipalo, K., Junninen, H., Mazon, S. B., Jokinen, T., Sarnela,
N., Sipilä, M., Petäjä, T., Kerminen, V.-M., Bianchi, F., and Kulmala, M.: Observations of biogenic
ion-induced cluster formation in the atmosphere, Sci. Adv., 4, 1–11, DOI:10.1126/sciadv.aar5218,
857  2018.

Schmale, J., Zieger, P., and Ekman, A. M. L.: Aerosols in current and future Arctic climate, Nat.
Clim. Change, 11, 95–105, https://doi.org/10.1038/s41558-020-00969-5, 2021.
Shiraiwa, M., Ueda, K., Pozzer, A., Lammel, G., Kampf, C. J., Fushimi, A., Enami, S., Arangio, A.
M., Fröhlich-Nowoisky, J., Fujitani, Y., Furuyama, A., Lakey, P. S. J., Lelieveld, J., Lucas, K.,
Morino, Y., Pöschl, U., Takahama, S., Takami, A., Tong, H., Weber, B., Yoshino, A., and Sato, K.:
Aerosol health effects from molecular to global scales, Environ. Sci. Technol., 51, 13545–13567,
https://doi.org/10.1021/acs.est.7b04417, 2017.
Shuman, N. S., Hunton, D. E., and Viggiano, A. A.: Ambient and modified atmospheric ion
chemistry: from top to bottom, Chem. Rev., 115, 4542–4570, https://doi.org/10.1021/cr5003479,
870  2015.

Sulo, J., Sarnela, N., Kontkanen, J., Ahonen, L., Paasonen, P., Laurila, T., Jokinen, T.,
Kangasluoma, J., Junninen, H., Sipilä, M., Petäjä, T., Kulmala, M., and Lehtipalo, K.: Long-term
measurement of sub-3 nm particles and their precursor gases in the boreal forest, Atmos. Chem.
Phys., 21, 695–715, https://doi.org/10.5194/acp-21-695-2021, 2021.





Tammet, H.: Size and mobility of nanometer particles, clusters and ions, J. Aerosol Sci., 26, 459–
878    475, 1995.
Tammet, H., Hõrrak, U., Laakso, L., and Kulmala, M.: Factors of air ion balance in a coniferous
forest according to measurements in Hyytiälä, Finland, Atmos. Chem. Phys., 6, 3377–3390,
https://doi.org/10.5194/acp-6-3377-2006, 2006.

Tammet, H, Komsaare, K., and Horrak, U.: Intermediate ions in the atmosphere, Atmos. Res., 135-
136, 263-273, https://doi.org/10.1016/j.atmosres.2012.09.009, 2014.

Tuovinen, S., Lampilahti, J., Kerminen, V.-M., and Kulmala, M.: Intermediate ions as indicator for
local new particle formation, Aerosol Research, 2, 93–105, https://doi.org/10.5194/ar-2-93-2024,
2024.

Wagner, R., Manninen, H. E., Franchin, A., Lehtipalo, K., Mirme, S., Steiner, G., Petäjä, T., and
Kulmala, M.: On the accuracy of ion measurements using a Neutral cluster and Air Ion
Spectrometer, Boreal Env. Res., 21, 230–241, 2016

Yan, C., Yin, R., Lu, Y., Dada, L., Yang, D., Fu, Y., Kontkanen,J., Deng, C., Garmash, O., Ruan, J.,
Baalbaki, R., Schervish, M., Cai, R., Bloss, M., Chan, T., Chen, T., Chen, Q., Chen, X., Chen, Y.,
Chu, B., Dällenbach, K., Foreback, B., He, X., Heikki-nen, L., Jokinen, T., Junninen, H.,
Kangasluoma, J., Kokkonen, T., Kurppa, M., Lehtipalo, K., Li, H., Li, H., Li, X., Liu, Y., Ma, Q.,
Paasonen, P., Rantala, P., Pileci, R. E., Rusanen, A., Sarnela, N., Simonen, P., Wang, S., Wang, W.,
Wang, Y., Xue, M., Yang, G., Yao, L., Zhou, Y., Kujansuu, J., Petäjä, T., Nie, W., Ma, Y., Ge, M.,
He, H., Donahue, N. M., Worsnop, D. R., Veli-Matti, K., Wang, L., Liu, Y., Zheng, J., Kulmala, M.,
Jiang, J., and Bianchi, F.: The Synergistic Role of Sulfuric Acid, Bases, and Oxidized Organics
Governing New-Particle Formation in Beijing, Geophys. Res. Lett., 48, e2020GL091944,
https://doi.org/10.1029/2020gl091944, 2021.

Yli-Juuti, T., Nieminen, T., Hirsikko, A., Aalto, P. P., Asmi, E., Hõrrak, U., Manninen, H. E.,
Patokoski, J., Dal Maso, M., Petäjä, T., Rinne, J., Kulmala, M., and Riipinen, I.: Growth rates of
nucleation mode particles in Hyytiälä during 2003−2009: variation with particle size, season, data
analysis method and ambient conditions, Atmos. Chem. Phys., 11, 12865–12886,
https://doi.org/10.5194/acp-11-12865-2011, 2011.


