# Peer review of "Investigating small ion number size distributions"

_Aerosol Research, 2025_

## Author Comment (AC1)

We thank both referees for their insightful comments. Please find the detailed answers below. The referee comments are marked in black, while the author comments are in turquoise. The changed parts from the improved manuscript are in *italics*.

**Referee 1:**

The authors present data from two sites with long-term observations of small ion number size distributions of both polarities, focusing on very small clusters of <1 nm to 2 nm in diameter. Complementary observations of sulfuric acid and organic molecular clusters are used to examine their impacts on the dynamics of cluster growth toward new particles. The two sites, Hyytiälä (remote forest) and Beijing (polluted urban), contrast in their environments and these contrasts are reflected in different dependencies of cluster dynamics on the measured precursors. These differences are illustrated with case study examples.

The data sets are interesting and unique, particularly in that they span multiple seasons and years. The introductory discussions of charged particles in the atmosphere and the charged general dynamic equation are well written and helpful in providing background for non-specialists. The observations are presented clearly, but the discussion and interpretation seemed less in-depth than suggested by the title "Insight into cluster formation and growth". Relationships are pointed to, but it's not entirely clear whether and how they shed new light into what is known already about cluster formation and growth, or how they might be applied to predict processes in other locations.

We thank the referee for seeing the potential in our analysis, and hope that the answers provided below will have improved these issues.

With the presentation and discussion of equation (2), I expected the data would be interpreted in that context, considering the different terms' influence on the evolution of the cluster sizes and particularly how the two sites support different magnitudes of some of the rates. I was therefore surprised that the analysis did not consider coagulation scavenging (CoagS). The discussion on lines 95-109 seems to indicate that this rate plays a key role, competing with the growth rate (GR) in shaping the small ion size distribution. Can the authors explain and justify why this term was not included in the analyses (Lines 108-109)? In particular, I anticipated that Beijing, as an urban region with higher particulate and precursor concentrations, would have lower overall ion concentrations due to competition for the ions by the coagulation sink. Indeed, Figure 2 shows that Beijing had lower number concentrations - but a rationale for this observation, supported by the data, is not provided here. On lines 342-344, the authors note that "Due to the high concentrations of both low volatility vapors and large particles, the dynamics of small ions in a megacity such as Beijing are different than in a rural site such as Hyytiälä." This, and other observations that note that the sites are "different", seems reasonable; but as an expected result, that isn't further explored with the dataset, it doesn't rise to the level of "insight".

We have now included the CoagS in the analysis. In addition to answering the question of how does the size distribution change with respect to CoagS, we hope it increases the insightfulness of our results.

Section 3.4 of the revised manuscript:

*"Fig. 7 shows the median small ion number size distributions in Hyytiälä and Beijing corresponding to the respective percentiles of condensation sink (CS). We see that the changes in the small ion size distribution with respect to changing CS are relatively small. In Hyytiälä (Fig. 7a, b), the*

*concentration of small ions, especially that of the smallest in diameter, decreases slightly with increasing CS. The sink is relatively low in Hyytiälä, and therefore this result is not unexpected.*

*Based on Fig. A1, which shows the approximate values of the different terms in Eq. 2, we would expect CoagS to have a much stronger impact on the small ion dynamics in Beijing than in Hyytiälä. The impact should be most clear for the smallest sizes. Surprisingly, this was not observed. Fig. 7c shows that in Beijing, the negative small ion size distribution below approx. 1.3 nm stays unchanged and the concentrations above decrease with an increasing CS. The larger positive small ion concentrations (Fig. 7d) also seem to slightly decrease with increasing sink, while the concentrations of the small ions close to 0.8 nm actually increase. Fig. A2a also shows that the total sub-2 nm concentration barely changes with changing CS. These results suggest that there could be a source of unknown nature for small ion cluster formation that is higher when CS is high, which would compensate for the increased coagulation scavenging of the ions.*

*Fig. A2d-f show the concentrations of the different low volatility vapors as a function of CS. We see that the concentration of sulfuric acid (Fig. A2d) and sulfuric acid dimers (Fig. A2e) decreases with increasing sink, as expected. However, the concentration of OOMs (Fig. A2f) increases with increasing sink. While this is purely speculation, if organic compounds are forming small ion clusters when the sink is higher, the weak apparent impact of CS on the small ion size distribution could be explained. Alternative potential explanation could be if there is a positive correlation between CS and the concentration of bases, which stabilize the small clusters. Regardless, the impact of CS on the statistics of the small ion size distribution in Beijing appears very small. "*

[Figure]

***Fig. 7 of the revised manuscript:*** *the median small ion size distributions grouped by the respective percentile of the  condensation sink (CS) values for Hyytiälä (a,b) and Beijing (c, d). The percentile value limits for CS in Hyytiälä are $1.3·10^{-3}$ $s^{-1}$, $2.3·10^{-3}$ $s^{-1}$, $3.5·10^{-3}$ $s^{-1}$, and $5.5·10^{-3}$ $s^{-1}$. In Beijing, the percentile value limits for CS are $9.3·10^{-3}$ $s^{-1}$, $1.9·10^{-2}$ $s^{-1}$, $3.1·10^{-2}$ $s^{-1}$, and $4.5·10^{-2}$ $s^{-1}$. "*

 We have also modified some of the existing discussion, taking into consideration the new results.

(L467):  "When looking at the small ion distributions in Beijing for different 2.0-2.3 nm ion concentrations or NPF ranking, unlike for low-volatility vapor concentrations, we are able to see the growth of small ions to intermediate ions in the size distribution. *These results show that the growth of small ions to larger diameters in Beijing is not limited by the availability of sulfuric acid or oxidized organic vapors, unlike in Hyytiälä. In addition, based on our analysis it does not appear to be strongly limited by CS either. This is supported by the relatively weak correlation between CS and the NPF ranking or 2.0-2.3 nm ions (Fig. A2b,c). Therefore, we speculate that the small ion growth could be limited more by the availability of bases. However, due to the lack of long-term base concentration data, this question remains unanswered.*"

(L610): "In Sect. 3.3, we saw how in Beijing there does not seem to be correlation between the small ion number size distribution and the concentration of sulfuric acid. *On this day, the increased*

*concentrations of sulfuric acid occurred approx. simultaneously with the observed small ion growth. Previous studies have shown the importance of sulfuric acid in particle formation in Beijing (Yao et al., 2018; Cai et al., 2021; Yan et al., 2021). As such, it seems likely that the growing small ions seen on this day are composed of sulfuric acid. However, while sulfuric acid forms these growing clusters, their growth also requires other ingredients"*

The main conclusions have also been modified to consider the applications of our results in future research (L649): *"The sub-2 nm size range is integral for understanding of the first steps of new particle formation and the activation of the clusters to grow into particles. Our results can be applied in research into the dynamics of charged clusters and how they grow from clusters to particles. "*

On line 421, the authors note: "In Beijing, CoagS is crucial in determining whether the growing clusters will survive to larger sizes or not." The subsequent section goes on to use new particle formation (NPF) ranking as a means of sorting for conditions supporting growth of cluster into larger sizes. If CoagS could not be determined explicitly and used as a sorting variable, but NPF ranking is a suitable proxy, this argument could be motivated and explained in the introduction. That would help explain more explicitly the reasons for the use of NPF ranking and would provide a clearer link to equation 2.

NPF ranking was used to find data from conditions that were suitable for cluster growth, not just with respect to CoagS but also to those parameters that we do not have data for (such as base concentrations), or have chosen to omit (such as meteorological data) from the analysis. We have now also added CoagS into the analysis and briefly clarified the role of the NPF ranking in the analysis

(L404): *We used the 2-2.3 nm ion concentrations and NPF ranking (Aliaga et al., 2023) as proxies for conditions that were favorable for cluster formation and growth.*

I suggest the authors also address two other points that might be helpful to make explicit. First, what do the ion polarities depend on? Are some molecules preferentially charged with one polarity over another and does composition play a role? The role of polarity in cluster dynamics and growth is not addressed extensively, so it wasn't clear why data were separated into positive and negative polarities for analysis and what should be gleaned from that, versus presenting the distributions of all charged ions together and how these totals trend with, e.g., sulfuric acid concentrations. Would different diurnal and seasonal trends in dominant polarity be expected, why, and what impacts on cluster dynamics are expected?

We have added (L145) to justify the division: *The chemical composition of small ions typically differs between the polarities (Ehn et al., 2010; Zha et al., 2023). For example, Ehn et al. (2010) found that in Hyytiälä the daytime negative small ions consisted largely of sulfuric acid clusters, while positive small ions consisted of organic species such as alkyl pyridines and alkyl amines. Therefore, we cannot assume that the negative and positive small ion populations behave similarly with respect to i.e., increased sulfuric acid concentrations. Thus, both negative and positive polarity were separately considered.*

Second, the role of transport and meteorology is not included explicitly in the discussion. The assumption seems to be that the observations are very localized (e.g., line 457) and can be interpreted as such; if this is the case, it would be helpful to explicitly state this and provide some justification. For example, is this approach justified because the lifetimes of these ions are so short?

Some of the observations do seem to suggest that impacts from other influences are seen in the data (e.g., line 491 mentions discontinuous growth in a Hyytiälä case study; Lines 539-540 suggest changes in ambient aerosol and meteorology affected the Beijing case study). If the behaviors are highly localized, what is their broader relevance?

*We have noted on the locality of the observations and the role of transport (L114): "We assume that the role of direct transport of clusters on the changes in the small ion size distribution is negligible due to their short lifetime of just a couple of minutes (Tammet et al., 2006). Therefore, the observations are assumed to be very local. However, transport can indirectly impact the size distribution of small ions i.e., through transport of trace gases and larger particles."*

*We have also added a small note on the exclusion of meteorology (L118): " We note that while meteorological conditions, such as temperature, strongly influence processes such as HOM formation (Quéléver et al., 2019), we do not explicitly consider them in this study."*

Overall, the data sets are very interesting and report characteristics of an understudied class of atmospheric particles. I suggest the manuscript and its utility for the community would be enhanced if the authors provide some additional interpretation (perhaps shortening the discussion that just points out trends, or move some to the Appendix), addressing the comments above.

Minor comments:

Line 328: "compared to results for Hyytiälä, the differences in the [Beijing] size distributions are small for either polarity." This refers to Figure 6a, compared with 5a or 3a or 3. This confused me, because in Figure 6a the shift in the peak diameter between positive and negative polarities looks larger than those in the other two figures.

*We have reformulated the sentence for clarity (L346): the differences in the size distributions with respect to different values of sulfuric acid or OOMs are small for both polarities.*

Line 81: should be "the changes IN"

Line 120: should be "from which data WERE"

Line 131: should be "WERE also used"

Line 171: should refer to TABLE 1 (not Table 2)

Line 197: should refer to TABLE 2 (not Table 3)

Line 290: note to author left in "add percentiles here"

Line 582: remains to be IDENTIFIED

*The above have now been corrected. We thank the referee for noticing them.*

**Referee 2:**

**General Comments**

The manuscript tackles a relevant question in atmospheric science: the dynamics of sub-2 nm ions and their interactions with precursor vapors and new particle formation (NPF). The study is well-motivated and supported by high-quality measurements from two contrasting environments (Hyytiälä and Beijing). The combination of long-term statistics with detailed case studies adds

significant value. However, the discussion falls short in some key areas—for example, the role of the coagulation sink and its implications for ion lifetime—and several technical and formatting issues affect clarity. Overall, the paper presents unique and valuable data and offers a clear view of ion behavior in different atmospheric settings, but the interpretation does not fully deliver the level of insight suggested by the title. Substantial revisions are needed to strengthen the analysis and improve coherence before publication.

We thank the referee for seeing the significance of our results and analysis, and hope that the answers provided below will have improved these issues.

**Specific Comments**

A major weakness concerns the treatment of the General Dynamic Equation (Eq. 2). Although correctly introduced, the analysis does not explore the influence of key terms such as the Coagulation Sink (CoagS), which is acknowledged as important in the text but explicitly excluded from the scope of this study. While this choice is understandable, a brief discussion of its potential impact on ion lifetime and survival—especially in high-PM environments like Beijing—would help contextualize the observed differences between sites. If feasible, even an approximate estimate could strengthen the interpretation, but this could also be addressed in future work.

Based on these comments, we have decided to include CoagS in the analysis.

Section 3.4 of the revised manuscript:

"Fig. 7 shows the median small ion number size distributions in Hyytiälä and Beijing corresponding to the respective percentiles of condensation sink (CS). We see that the changes in the small ion size distribution with respect to changing CS are relatively small. In Hyytiälä (Fig. 7a, b), the concentration of small ions, especially that of the smallest in diameter, decreases slightly with increasing CS. The sink is relatively low in Hyytiälä, and therefore this result is not unexpected.

Based on Fig. A1, which shows the approximate values of the different terms in Eq. 2, we would expect CoagS to have a much stronger impact on the small ion dynamics in Beijing than in Hyytiälä. The impact should be most clear for the smallest sizes. Surprisingly, this was not observed. Fig. 7c shows that in Beijing, the negative small ion size distribution below approx. 1.3 nm stays unchanged and the concentrations above decrease with an increasing CS. The larger positive small ion concentrations (Fig. 7d) also seem to slightly decrease with increasing sink, while the concentrations of the small ions close to 0.8 nm actually increase. Fig. A2a also shows that the total sub-2 nm concentration barely changes with changing CS. These results suggest that there could be a source of unknown nature for small ion cluster formation that is higher when CS is high, which would compensate for the increased coagulation scavenging of the ions.

Fig. A2d-f show the concentrations of the different low volatility vapors as a function of CS. We see that the concentration of sulfuric acid (Fig. A2d) and sulfuric acid dimers (Fig. A2e) decreases with increasing sink, as expected. However, the concentration of OOMs (Fig. A2f) increases with increasing sink. While this is purely speculation, if organic compounds are forming small ion clusters when the sink is higher, the weak apparent impact of CS on the small ion size distribution could be explained. Alternative potential explanation could be if there is a positive correlation between CS and the concentration of bases, which stabilize the small clusters. Regardless, the impact of CS on the statistics of the small ion size distribution in Beijing appears very small.

[Figure]

**Fig. 7 of the revised manuscript:** *the median small ion size distributions grouped by the respective percentile of the condensation sink (CS) values for Hyytiälä (a,b) and Beijing (c, d). The percentile value limits for CS in Hyytiälä are $1.3 \cdot 10^{-3}$ $s^{-1}$, $2.3 \cdot 10^{-3}$ $s^{-1}$, $3.5 \cdot 10^{-3}$ $s^{-1}$, and $5.5 \cdot 10^{-3}$ $s^{-1}$. In Beijing, the percentile value limits for CS are $9.3 \cdot 10^{-3}$ $s^{-1}$, $1.9 \cdot 10^{-2}$ $s^{-1}$, $3.1 \cdot 10^{-2}$ $s^{-1}$, and $4.5 \cdot 10^{-2}$ $s^{-1}$. "*

The discussion on polarity differences (negative vs. positive ions) is too brief. Since the data are presented separately for each polarity, a more detailed mechanistic explanation of why polarity matters—e.g., whether precursor composition influences charge preference—would improve clarity.

We have added (L145) to justify the division: *The chemical composition of small ions typically differs between the polarities (Ehn et al., 2010; Zha et al., 2023). For example, Ehn et al. (2010) found that in Hyytiälä the daytime negative small ions consisted largely of sulfuric acid clusters, while positive small ions consisted of organic species such as alkyl pyridines and alkyl amines. Therefore, we cannot assume that the negative and positive small ion populations behave similarly with respect to i.e., increased sulfuric acid concentrations. Thus, both negative and positive polarity were separately considered.*

Growth rates (GR) are reported for selected case studies, which is consistent with the stated scope. However, the manuscript would benefit from clarifying why a systematic analysis across the full

dataset was not attempted. Providing uncertainty estimates for the reported GR values, or at least acknowledging their variability, would enhance transparency.

Systematic analysis of GRs for the full dataset is out of the scope of this manuscript. The purpose of GRs is to support the analysis of the three example days. We have now noted that the values are for that particular example case, and not meant to be applied more widely to the whole dataset (L524): "[on GR in Case 1] Regardless, it shows that the growth of ions below 2.0 nm is non-negligible *on this particular day. We note that this GR, or the ones presented for Hyytiälä Case 2 and Beijing Case, is not a representative of the whole range of GRs for similar cases in the same location, and that there can be considerable variability."*

and (L566): "[on GR in Case 2] The negative ion GR between 1.43 to 2.05 nm was estimated to be 1.28 nm/h (Fig. A6), which is over twice as high as the GR estimated for Case 1, likely due to the high concentration of lower volatility HOMs driving the small ion growth *during this particular evening. "*

Additional context is missing: meteorological factors such as temperature and relative humidity, which strongly affect seasonal variability and processes like HOM formation, are not considered. While this omission is consistent with the study's focus, a short statement acknowledging these influences would help readers interpret the results and understand limitations.

We have added the following sentence to address this (L118): *"We note that while meteorological conditions, such as temperature, strongly influence processes such as HOM formation (Quéléver et al., 2019), we do not explicitly consider them in this study."*

Two coherence issues require attention.

The discrepancy between weak correlations in Beijing (Section 3.3) and clear growth observed in case studies should be explicitly clarified

This is addressed in (L610): "In Sect. 3.3, we saw how in Beijing there does not seem to be correlation between the small ion number size distribution and the concentration of sulfuric acid. *On this day, the increased concentrations of sulfuric acid occurred approx. simultaneously with the observed small ion growth. Previous studies have shown the importance of sulfuric acid in particle formation in Beijing (Yao et al., 2018; Cai et al., 2021; Yan et al., 2021). As such, it seems likely that the growing small ions seen on this day are composed of sulfuric acid. However, while sulfuric acid forms these growing clusters, their growth also requires other ingredients"*

Implications for atmospheric modelling and climate feedbacks, mentioned in the introduction, are not revisited in the conclusions. Even a brief statement on how these findings could inform models would improve the manuscript's impact.

We have modified the main conclusions of the paper to consider the application of our results (L649): *"The sub-2 nm size range is integral for understanding of the first steps of new particle formation and the activation of the clusters to grow into particles. Our results can be applied in research into the dynamics of charged clusters and how they grow from clusters to particles."*

**Technical Comments**

**Pag.2, line 57** "…. complex climate-biosphere feedbacks (Kulmala et al., 2020; Kulmala et al., Kulmala et al., 2024b)." Reference might be missing or incorrectly formatted.

The incorrect formatting has been corrected.

**Pag. 3, line 81** "The changing in the small ion number concentration can be described…" change in: "The change in the small ion number concentration can be described…"

This has been changed as proposed.

**Pag. 5, line 171**

The text refers to Table 2 for Hyytiälä data, but the correct table is Table 1.

Example suggestion: Replace "Table 2" with "Table 1".

Fixed.

**Pag. 5, line 180** "…behave similarly as the negative ones…" change in: "…behave similarly to the negative ones…"

This has been changed as proposed.

**Pag. 6, line 197**

The text refers to Table 3, which does not exist; the correct reference is Table 2.

Fixed.

**Pag. 11, line 296** "…organic compounds tend drive cluster growth…"change to: "…organic compounds tend to drive cluster growth…"

Fixed.

**Pag. 14, line 362** "…we can see from the small ion size distribution for both polarities how this growth of small ions up to 2.0 nm is seen…" change in: "…the small ion size distribution for both polarities shows this growth of small ions up to 2.0 nm…"

Changed as proposed

**Pag. 19, line 465** "…over a order of magnitude higher…" change in: "…over an order of magnitude higher…"

Fixed.

**Fig. 1**

X-axis scales differ between panels for Hyytiälä and Beijing, making comparison difficult.

Example suggestion: Use a common X-axis scale for both panels.

**Fig. 3, Fig. 5, Fig. 6, Fig. 7, Fig. 9, Fig. 10–12**

The text refers to panels (a, b, c, d), but the figures do not display these labels.

Example suggestion: Add panel labels (a, b, etc.) to all multi-panel figures for consistency.

**Fig. 5**

Missing panel labels and inconsistent panel sizes: the bottom-right scatter plot is smaller than the top-right panel.

Example suggestion: Add panel labels and make panel sizes uniform.

**All figures with time ranges (e.g., Fig. 3, Fig. 6, Fig. 9, Fig. 10–12)**

Time formatting is inconsistent: "8–16:00", "8:00–16:00", "08:00–16:00".

Example suggestion: Use a consistent format such as "08:00–16:00".

These above-mentioned issues in the figures have been fixed as suggested by the referee.